# Activation of transient receptor potential vanilloid 4 is involved in pressure overload-induced cardiac hypertrophy

**Yan Zou[1,2†], Miaomiao Zhang[1†], Qiongfeng Wu[3], Ning Zhao[3], Minwei Chen[4], Cui Yang[4], Yimei Du[3]\*, Bing Han[1]\***

[1]Department of Cardiology, Xuzhou Central Hospital, Xuzhou, China; [2]Xuzhou Institute of Cardiovascular Disease, Xuzhou Central Hospital, Xuzhou, China; [3]Department of Cardiology, Union Hospital, Tongji Medical College, Huazhong University of Science and Technology, Wuhan, China; [4]Department of Cardiology, Xiamen Key Laboratory of Cardiac Electrophysiology, Xiamen Institute of Cardiovascular Diseases, The First Affiliated Hospital of Xiamen University, School of Medicine, Xiamen University, Xiamen, China

**\*For correspondence:**
yimeidu@mail.hust.edu.cn (YD);
hbing777@hotmail.com (BH)

†These authors contributed equally to this work

**Competing interest:** The authors declare that no competing interests exist.

**Abstract** Previous studies, including our own, have demonstrated that transient receptor potential vanilloid 4 (TRPV4) is expressed in hearts and implicated in cardiac remodeling and dysfunction. However, the effects of TRPV4 on pressure overload-induced cardiac hypertrophy remain unclear. In this study, we found that TRPV4 expression was significantly increased in mouse hypertrophic hearts, human failing hearts, and neurohormone-induced hypertrophic cardio-myocytes. Deletion of TRPV4 attenuated transverse aortic constriction (TAC)-induced cardiac hypertrophy, cardiac dysfunction, fibrosis, inflammation, and the activation of NFκB - NOD - like receptor pyrin domain-containing protein 3 (NLRP3) in mice. Furthermore, the TRPV4 antagonist GSK2193874 (GSK3874) inhibited cardiac remodeling and dysfunction induced by TAC. In vitro, pretreatment with GSK3874 reduced the neurohormone-induced cardiomyocyte hypertrophy and intracellular $Ca^{2+}$ concentration elevation. The specific TRPV4 agonist GSK1016790A (GSK790A) triggered $Ca^{2+}$ influx and evoked the phosphorylation of $Ca^{2+}$/calmodulin-dependent protein kinase II (CaMKII). But these effects were abolished by removing extracellular $Ca^{2+}$ or GSK3874. More importantly, TAC or neurohormone stimulation-induced CaMKII phosphorylation was significantly blocked by TRPV4 inhibition. Finally, we show that CaMKII inhibition significantly prevented the phosphorylation of NFκB induced by GSK790A. Our results suggest that TRPV4 activation contributes to pressure overload-induced cardiac hypertrophy and dysfunction. This effect is associated with upregulated $Ca^{2+}$/CaMKII mediated activation of NFκB-NLRP3. Thus, TRPV4 may represent a potential therapeutic drug target for cardiac hypertrophy and dysfunction after pressure overload.

## Editor's evaluation

Zou et al., demonstrate that activation of the TRPV4 ion channel is a novel contributor to cardiac hypertrophy, adding to the number of mechanisms known to trigger pathological left ventricular hypertrophy (LVH), While the exact signaling pathway downstream of TRPV4 leading to pathological hypertrophy still remain to be uncovered, the findings that this ion channel contributes to cardiac hypertrophy and that the Piezo1 channel acts upstream of PLA2 and TRPV4 add to our understanding of the pathogenesis of LVH and heart failure.

## Introduction

In response to pathological stimuli such as hypertension, valvular heart disease, and neurohumoral overactivation, the heart undergoes hypertrophy. Initially, the hypertrophy response is adaptive, yet sustained cardiac hypertrophy results in increased heart mass, cardiac fibrosis, and eventually heart failure (*Bui et al., 2011*; *Nakamura and Sadoshima, 2018*). Although significant advances in the treatment of pathological hypertrophy, heart failure is still a leading cause of death worldwide (*Neubauer, 2007*). Thus, deeply uncovering the molecular mechanism of pathological cardiac hypertrophy continues to be important for developing novel therapeutic strategies for the prevention of cardiac remodeling and dysfunction (*Kalman et al., 2019*).

Increased mechanical stress plays a key role in cardiac hypertrophy. The transient receptor potential vanilloid (TRPV) channels are ubiquitous ion channels that function as essential mechanical sensors (*Clapham, 2003*). Interestingly, those channels are upregulated in the hearts of mice after transverse aortic constriction (TAC), as shown for TRPV1, TRPV2, and TRPV3 (*Chen et al., 2016*; *Zhang et al., 2018*). Furthermore, the genetic deletion of functional TRPV2 significantly ameliorates TAC-induced cardiac hypertrophy and dysfunction (*Koch et al., 2017*). These findings suggest that the role of TRPV is critical in the development of cardiac remodeling in response to pressure overload.

TRPV4, a member of the TRPV subfamily, is widely expressed in the cardiovascular system (*Hof et al., 2019*; *White et al., 2016*). Its functional expression is increased under certain pathological conditions, such as pressure overload (*Morine et al., 2016*), aging (*Jones et al., 2019*), ischemia-reperfusion (*Dong et al., 2017*; *Wu et al., 2017b*), and pericarditis (*Liao et al., 2020*). Inhibition of TRPV4 attenuates intracellular calcium concentration ($[Ca^{2+}]_i$) (*Wu et al., 2017b*), cardiac fibrosis (Adapala, et al.,2020), and cardiac inflammation (*Liao et al., 2020*), which improves cardiac function (*Wu et al., 2019*). In addition, a potent and selective TRPV4 inhibitor recently revealed a positive efficacy trend in a Phase 2a trial in patients with heart failure (*Goyal et al., 2019*; *Stewart et al., 2020*). Moreover, Adapala et al. have reported that endothelial TRPV4 deletion protects TAC-induced-structure remodeling in a conference abstract (*Adapala et al., 2019*). However, the role of TRPV4 in the development of pressure overload-induced cardiac hypertrophy is not well understood. Therefore, in the present study, we aim to investigate the role and the underlying mechanism of TRPV4 in pathological cardiac hypertrophy subjected to pressure overload.

## Results

### TRPV4 expression is increased in pathological cardiac hypertrophy

To evaluate the potential role of TRPV4 in cardiac hypertrophy, we first measured TRPV4 protein and mRNA expression levels in left ventricle (LV) tissue from wild-type (WT) TAC versus sham mice at different time points (2, 7, 14, and 28 days) after surgery. Consistent with the previous observation (*Guo et al., 2021b*), we did not detect significant cardiac hypertrophy on day 2 after TAC. The 1 week TAC time point gave a sample in which the heart was undergoing compensated cardiac hypertrophy, while the 2- and 4-week TAC showed signs of decompensated cardiac hypertrophy with heart failure (*Appendix 1—table 1*). Similar findings have been previously reported (*Wu et al., 2017a*). Nonetheless, there are conflicting results for changes in cardiac function, particularly 2 weeks after TAC, likely caused by differences in the severity of constriction. As shown in *Figure 1A–C* and *Source data 1*, the protein and mRNA level of TRPV4 began to increase 1 week after TAC and maintained a higher level on week 4. We also assessed the TRPV4 expression level in LV tissue from human hearts and found that TRPV4 protein was significantly upregulated in failing hearts compared with non-failing (*Figure 1D–E*, *Source data 1*). Our results indicate that TRPV4 may be implicated in pathological cardiac hypertrophy and heart failure.

### TRPV4 deficiency attenuates cardiac hypertrophy induced by pressure overload in vivo

To further investigate the role of TRPV4 in cardiac hypertrophy induced by pressure overload, we performed TAC or sham surgery in WT and TRPV4 knockout (*Trpv* KO) mice. The hypertrophic response was evaluated 1 and 4 weeks after TAC. We used the ratios of heart weight/body weight (HW/BW) and HW/tibial length (TL) to assess changes in LV mass (*Figure 2A*). As expected, both values were significantly increased after TAC in WT mice. However, this hypertrophic response to

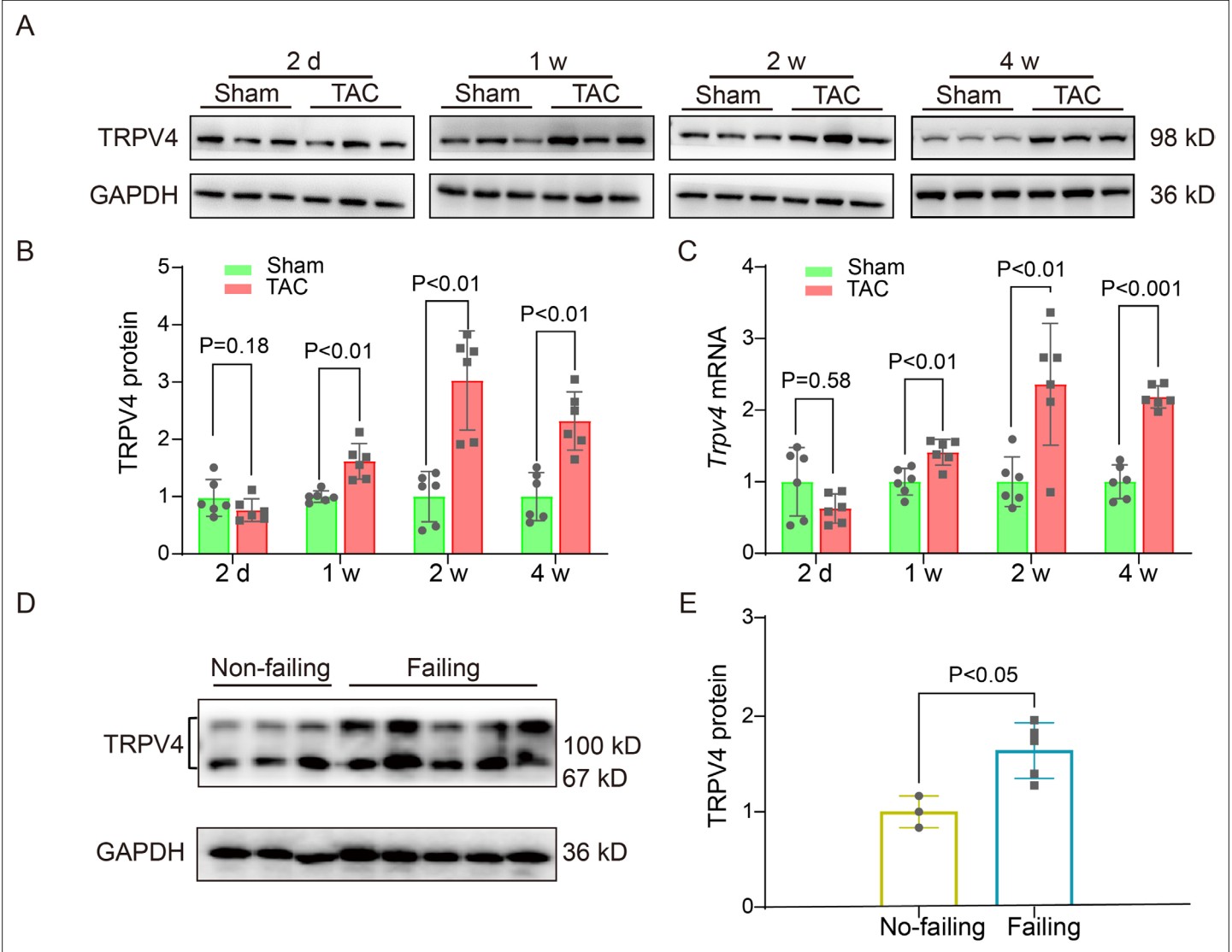

**Figure 1.** TRPV4 expression is upregulated in pathological cardiac hypertrophy. Representative immunoblot image (**A**) and statistics (**B**) of TRPV4 protein level in the LV from sham or TAC mice at indicated time points after the operation (n=6 per group). All results represent mean ± SD, an unpaired two-tailed Student's t-test. (**C**) Statistical data of *Trpv4* mRNA level in the LV from sham or TAC mice at indicated time points after the operation (n=6 per group). All results represent mean ± SD, an unpaired two-tailed Student's t-test. Representative immunoblot image (**D**) and statistical data (**E**) of TRPV4 protein level in human non-failing hearts (n=3) and failing hearts (n=5). All results represent mean ± SD, an unpaired two-tailed Student's t-test. LV, left ventricle; TAC, transverse aortic constriction.

The online version of this article includes the following source data for figure 1:

**Source data 1.** Source data file (Excel) for *Figure 1B*.

**Source data 2.** Source data file (Excel) for *Figure 1C*.

**Source data 3.** Source data file (Excel) for *Figure 1E*.

TAC was attenuated in *Trpv4* KO mice at 1 or 4 weeks. Next, we measured the cross-sectional area of myocytes in all groups. As shown in **Figure 2B**, *Trpv4* KO mice significantly attenuated TAC-induced enlargement of myocytes size 1 and 4 weeks after TAC. In order to confirm our findings at the molecular level, we then determined cardiac hypertrophic marker genes expression. Both ANP (*Nppa*) and BNP (*Nppb*) mRNA expression were significantly higher in WT hearts compared with *Trpv4* KO hearts 1 and 4 weeks after TAC. There was no significant difference between WT and *Trpv4* KO in the sham group (**Figure 2C**). These results suggest that TRPV4 activation plays a critical role in pressure overload-induced cardiac hypertrophy.

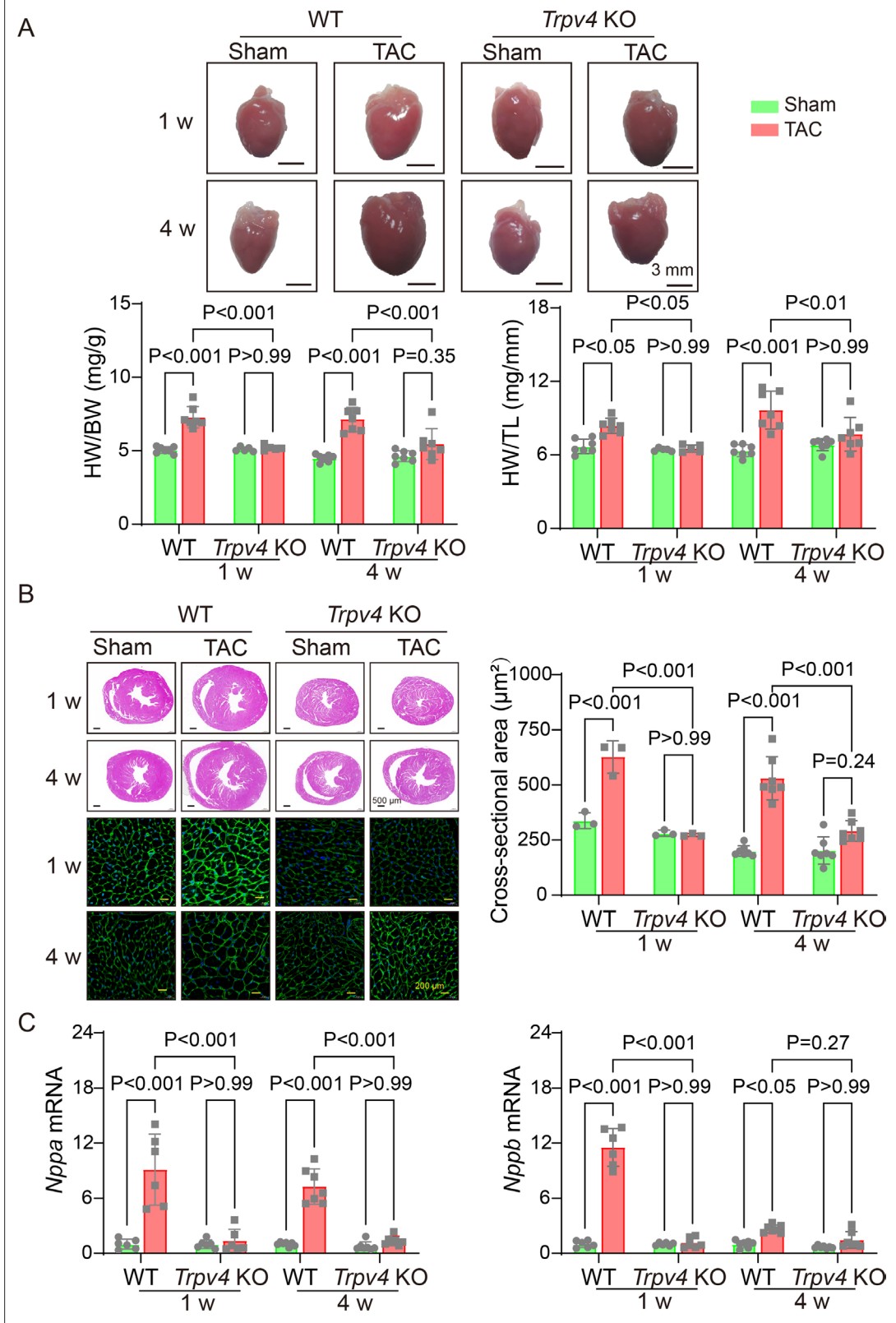

**Figure 2.** TRPV4 deficiency attenuates pressure overload-induced cardiac hypertrophy. (**A**) Representative images of the heart and statistical results for the ratios of HW/BW and HW/TL of WT and *Trpv4* KO mice 1 or 4 weeks after sham or TAC operation (n=7 per group in WT mice, n=5 per group at 1 week in TRPV4⁻/⁻ mice, n=7 per group at 4 weeks in *Trpv4* KO mice) (**B**) Hematoxylin & eosin staining, wheat germ agglutinin staining, and cross-section area in WT and *Trpv4* KO mice 1 or 4 weeks after sham or TAC operation (n=3 per group at 1 week, n=7 per group at 4 weeks). (**C**) Statistics

*Figure 2 continued on next page*

*Figure 2 continued*

of hypertrophy-related genes ANP (*Nppa*) (**E**) and BNP (*Nppb*) (**F**) mRNA levels in mouse hearts from WT or *Trpv4* KO 1 or 4 weeks after sham or TAC operation (n=6 per group at 1 week, n=7 per group at 4 weeks). All results represent mean ± SD, a two-way ANOVA followed by the Bonferroni test. BW, body weight; HW, heart weight; TAC, transverse aortic constriction; TL, tibial length; WT, wild-type.

The online version of this article includes the following source data for figure 2:

**Source data 1.** Source data file (Excel) for *Figure 2A, B and C*.

## TRPV4 deficiency attenuates cardiac dysfunction and cardiac fibrosis induced by pressure overload

Echocardiography was performed to monitor the progression of cardiac structure and functional changes (*Figure 3A*). A reduction in ejection fraction (EF, 52.83±10.62% vs. 73.44±6.05%, p<0.001, *Figure 3B*) and fractional shortening (FS, 26.87±6.46% vs. 41.34±4.83%, p<0.001, *Figure 3C*) in WT mice were reversed in *Trpv4* KO mice at 4 weeks after TAC. Consistently, LV internal dimension systole and LV mass were significantly increased in WT TAC mice, but these effects were not found in *Trpv4* KO TAC mice (*Figure 3D–E*). Other parameters of LV remodeling, including LV posterior end-diastolic wall thickness (LVPW), LV end-diastolic diameter (LVEDD), and LV end-diastolic volume (LVEDV), were also well preserved in *Trpv4* KO mice compared with WT mice after TAC (*Table 1*).

Cardiac interstitial and perivascular fibrosis were assessed in Masson's Trichrome stained sections 4 weeks after TAC surgery (*Figure 3F*). There was no significant difference in the extent of fibrosis in WT and *Trpv4* KO mice in the sham groups. However, both interstitial and perivascular fibrosis increased in WT hearts after TAC, with more pronounced perivascular changes. The increase in interstitial and perivascular fibrosis was significantly blunted in *Trpv4* KO hearts after TAC (2.48±0.95% vs.

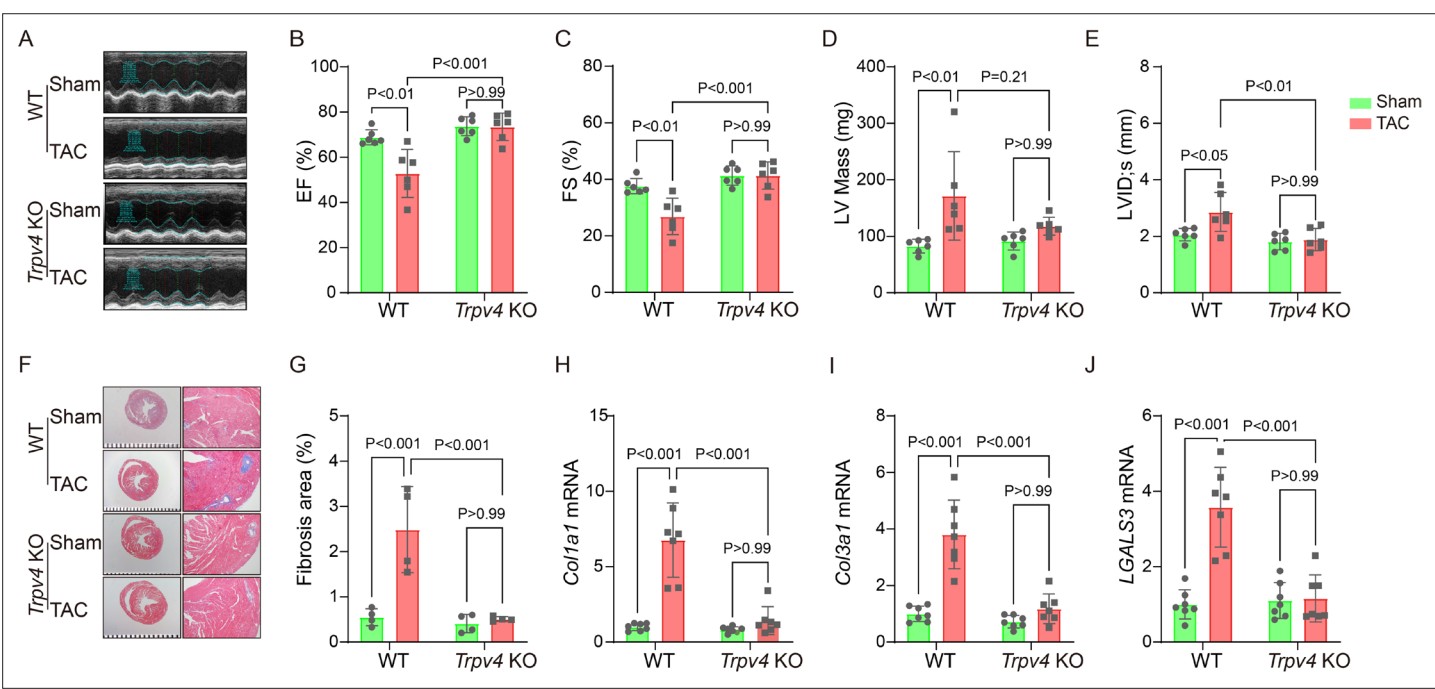

**Figure 3.** TRPV4 deficiency improves cardiac function and attenuates cardiac fibrosis induced by pressure overload. Representative images of M-mode echocardiography of WT and *Trpv4* KO mice 4 weeks after sham or TAC operation (**A**). Statistics of EF (**B**), FS (**C**), LV mass (**D**), and LVIDs (**E**) in mice 4 weeks after sham or TAC operation (n=6 per group). Representative images (**F**) and statistics (**G**) of Masson's trichrome-stained hearts from mice 4 weeks after sham or TAC operation. The statistics were from the panoramic scanning pictures (n=4 per group). Statistics of fibrosis-related genes collagenase-1 (*Col1a1*) (**H**), collagenase-3 (*Col1a1*) (**I**), and galectin-3 (*LGALS3*) (**J**) mRNA levels in mouse hearts 4 weeks after sham or TAC operation (n=7 per group). All results represent mean ± SD, a two-way ANOVA followed by the Bonferroni test. EF, ejection fraction; FS, fractional shortening; LV, left ventricle; TAC, transverse aortic constriction.

The online version of this article includes the following source data for figure 3:

**Source data 1.** Source data file (Excel) for *Figure 3B, C, D, E, G, H, I*.

**Table 1.** Echocardiographic measurements 4 weeks after TAC.

| | WT | | Trpv4 KO | |
|---|---|---|---|---|
| | Sham | TAC | Sham | TAC |
| Heart rate (bpm) | 457.61±11.77 | 463.11±16.96 | 463.11±18.86 | 443.78±10.38 |
| LVAW,s (mm) | 1.29±0.14 | 1.41±0.18 | 1.21±0.2 | 1.48±0.17 |
| LVID,d (mm) | 3.32±0.12 | 3.81±0.59 | 2.92±0.38 | 3.11±0.47 |
| LVPW,d (mm) | 0.70±0.08 | 1.08±0.46 | 0.92±0.39 | 1.02±0.37 |
| LVPW,s (mm) | 1.17±0.11 | 1.41±0.57 | 1.33±0.34 | 1.59±0.31 |
| Diameter,s (mm) | 2.05±0.13 | 2.67±0.57 | 1.69±0.32 | 1.80±0.37## |
| Diameter,d (mm) | 3.29±0.12 | 3.62±0.48 | 2.87±0.42 | 3.04±0.45 |
| Volume,s (µl) | 13.73±2.17 | 28.01±14.43* | 8.79±4.00 | 10.36±5.41## |
| Volume,d (µl) | 43.88±3.84 | 56.39±17.41 | 32.23±11.02 | 37.38±12.30 |
| Stroke volume (µl) | 30.14±2.43 | 28.39±4.67 | 23.44±7.19 | 27.02±7.52 |

LVAW =s: systolic left ventricular anterior wall = LVID,s: systolic left ventricular internal diameter = LVPW,d: diastolic left ventricular posterior wall = LVPW,s.: systolic left ventricular posterior wall = Data represent means ± SD, n=6 per group, *$P<0.05$ WT TAC vs WT sham group. ## $P<0.01$ Trpv4 KO TAC vs WT TAC group

The online version of this article includes the following source data for table 1:

**Source data 1.** Source data file (Excel) for *Table 1*.

0.51±0.05%, p<0.001, *Figure 3G*). In addition, quantitative real-time PCR revealed a marked reduction in fibrosis markers (collagenase-1, collagenase-3, and galectin-3, *Figure 3H–J*).

## TRPV4 deficiency attenuates the inflammation induced by pressure overload

Chronic inflammation promotes cardiac fibrosis (*Adamo et al., 2020*). Thus, we detected the protein and mRNA levels of pro-inflammatory cytokines 4 weeks after TAC. As shown in *Figure 4A–D* and *Source data 1*, TAC significantly upregulated the protein levels of IL-1β, IL-6, and TNF-α in WT mice, and TRPV4 deletion diminished this elevation. Consistent with these observations, the TAC-induced increases in mRNA expression of IL-1β (*Il1b*), IL-6 (*Il6*), TNF-α (*Tnfa*), MIP-2 (*Mip2*), and MCP-1 (*Mcp1p*) were significantly attenuated in *Trpv4* KO mice (*Figure 4E–I*).

The NOD-like receptor pyrin domain-containing protein 3 (NLRP3) inflammasome consists of ASC, NLRP3, and caspase-1 (*Martinon and Tschopp, 2004*). Its activation contributes to the development of cardiac hypertrophy by cleaving pro-caspase-1 and promoting the release of proinflammatory cytokine IL-β (*Suetomi et al., 2019*; *Suetomi et al., 2018*). NFκB represents a family of inducible transcription factors, which regulate various genes involved in inflammatory responses. We then assessed the activation of the NLRP3 inflammasome and the phosphorylation of NFκB (*Figure 4J*, *Source data 1*). As shown in *Figure 4K–L*, TAC significantly upregulated the protein levels of ASC, NLRP3, and cleaved caspase-1 (p20) in WT mice. We also found that the expression of p-NFκB p65 (ser536) was greatly upregulated in WT mice after TAC surgery (*Figure 4N*). Interestingly, TRPV4 deletion efficiently reduced the ASC, NLRP3, cleaved caspase-1, and p-NFκB p65 protein levels.

## TRPV4 antagonist attenuates the pathological cardiac remodeling induced by TAC

We further evaluated the effects of TRPV4 inhibition with a specific antagonist GSK2193874 (GSK3874) on pathological cardiac hypertrophy and dysfunction induced by pressure overload. WT mice received intragastric administration of GSK3874 at a dose of 10 mg/kg/d or vehicle from the day of TAC operation (*Thorneloe et al., 2012*; *Liao et al., 2020*). After 4 weeks, GSK3874 treatment substantially suppressed the increase in heart size induced by TAC (*Figure 5A*). Moreover, HW/BW and HW/TL ratios significantly decreased in GSK3874-treated mice (*Figure 5B–C*). The cross-sectional area of cardiomyocytes was also markedly reduced after treatment with GSK3874 (*Figure 5D*). The mRNA expression

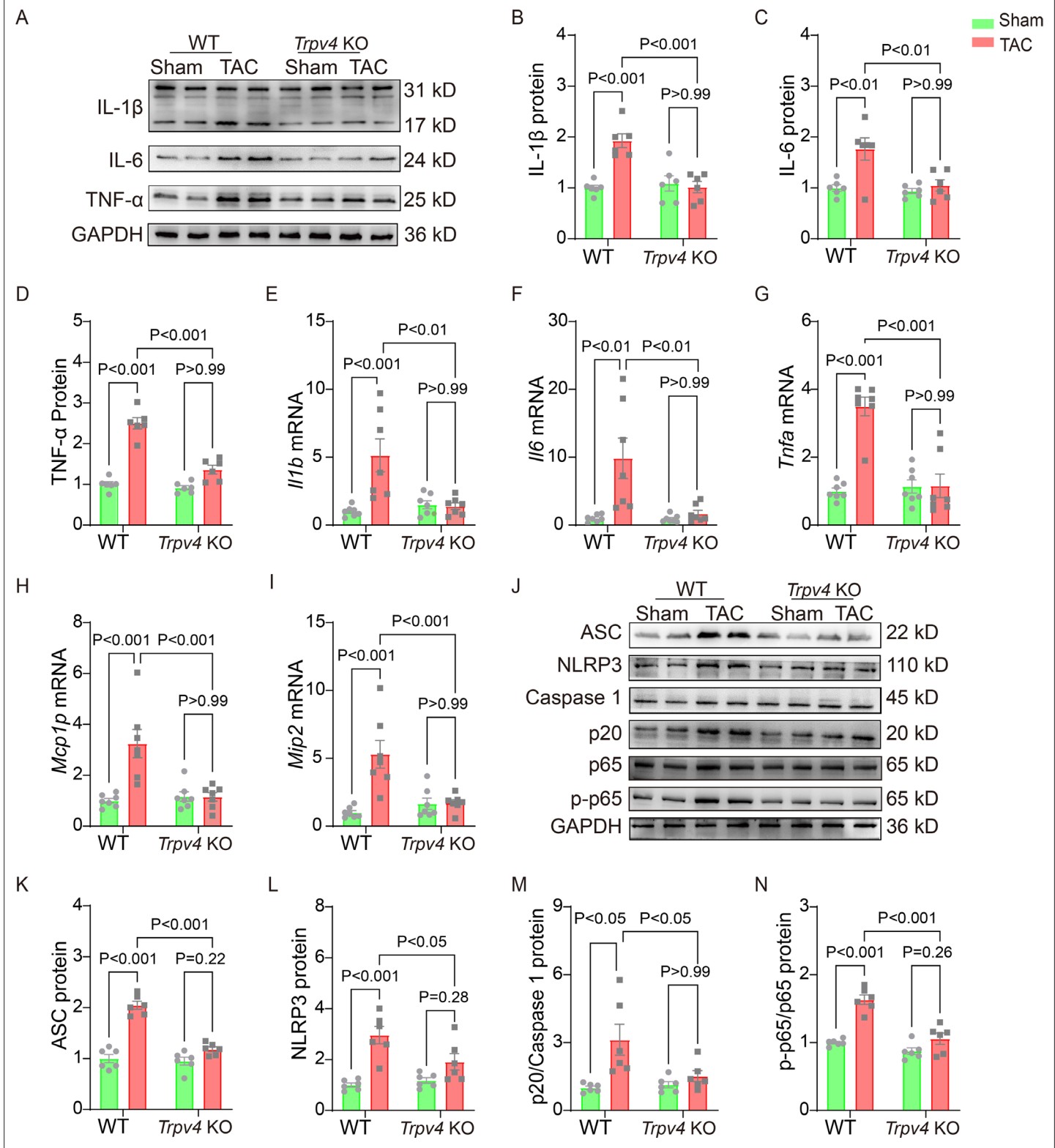

**Figure 4.** TRPV4 deficiency attenuates cardiac fibrosis induced by pressure overload. Representative immunoblot image (**A**) and statistics of IL-1β (**B**), IL-6 (**C**), and TNF-α (**D**) protein levels in WT and *Trpv4* KO mice 4 weeks after sham or TAC operation (n=6 per group). Statistical data of IL-1β (*Il1b*) (**E**), IL-6 (*Il6*) (**F**), TNF-α (*Tnfa*) (**G**), MCP-1 (*Mcp1p*) (**H**), and MIP-2 (*Mip2*) (**I**) mRNA levels in mouse hearts 4 weeks after sham or TAC operation (n=7 per group). Representative immunoblot image (**J**) and statistics of ASC (**K**), NLRP3 (**L**), Caspase 1-p20 (**M**), and p-NF κ B p65 (**N**) protein levels in WT and TRPV4⁻/⁻

*Figure 4 continued on next page*

*Figure 4 continued*

mice at 4 weeks after sham or TAC operation (n=6 per group). All results represent mean ± SD, a two-way ANOVA followed by the Bonferroni test. TAC, transverse aortic constriction; WT, wild-type.

The online version of this article includes the following source data for figure 4:

**Source data 1.** Source data file (Excel) for *Figure 4B, C and D*.

**Source data 2.** Source data file (Excel) for *Figure 4E-I*.

**Source data 3.** Source data file (Excel) for *Figure 4K, L, M and N*.

of hypertrophy marker genes ANP *(Nppa)* and BNP *(Nppb)* was significantly reduced (*Figure 5E–F*). Echocardiography showed that EF% and FS% were also significantly higher in GSK3874- than in the vehicle-treated group (*Figure 5G–H*). LV internal dimension systole and LV mass were significantly smaller following GSK3874 treatment (*Figure 5I–J*). Finally, the fibrosis area significantly decreased in

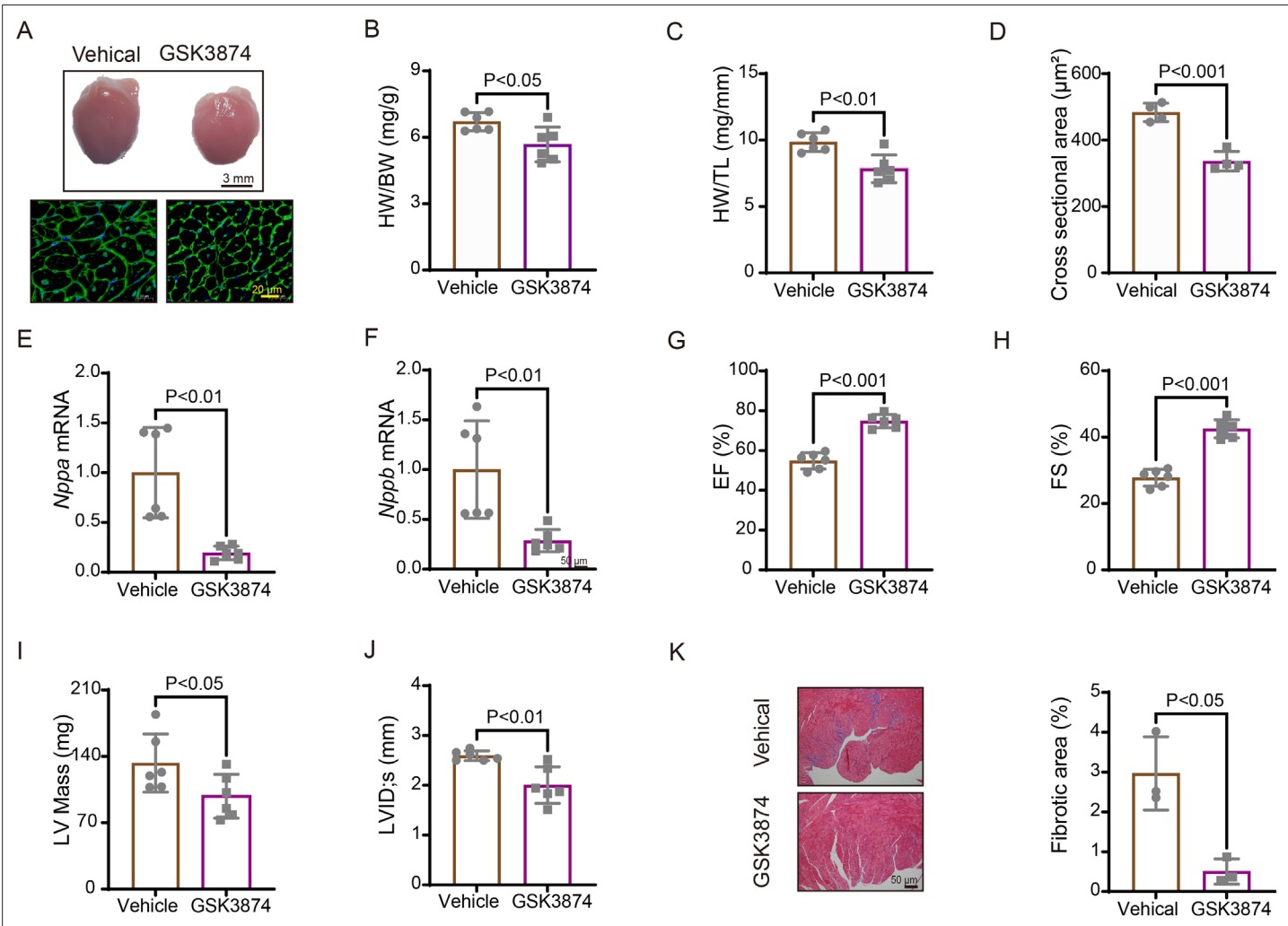

**Figure 5.** Treatment with TRPV4 antagonist prevents TAC-induced cardiac hypertrophy and dysfunction in mice. (**A**) Representative images of heart photo and WGA staining 4 weeks after TAC. Statistical results for HW/BW ratio (n=6 per group) (**B**), HW/TL ratio (n=6 per group) (**C**), cross-section area (n=4 per group) (**J**), ANP *(Nppa)* (n=6 per group) (**E**), BNP *(Nppb)* (n=6 per group) (**F**), EF (n=6 per group) (**G**), FS (n=6 per group) (**H**), LV mass (n=6 per group) (**I**), and LVIDs (n=6 per group) (**J**). All results represent mean± SD, an unpaired two-tailed Student's t-test. Representative images and statistics (**K**) of Masson's trichrome-stained hearts 4 weeks after TAC (n=3 per group). All results represent mean ± SD, an unpaired two-tailed Student's t-test. EF, ejection fraction; FS, fractional shortening; LV, left ventricle; TAC, transverse aortic constriction; WGA, wheat germ agglutinin.

The online version of this article includes the following source data for figure 5:

**Source data 1.** Source data file (Excel) for *Figure 5C–K*.

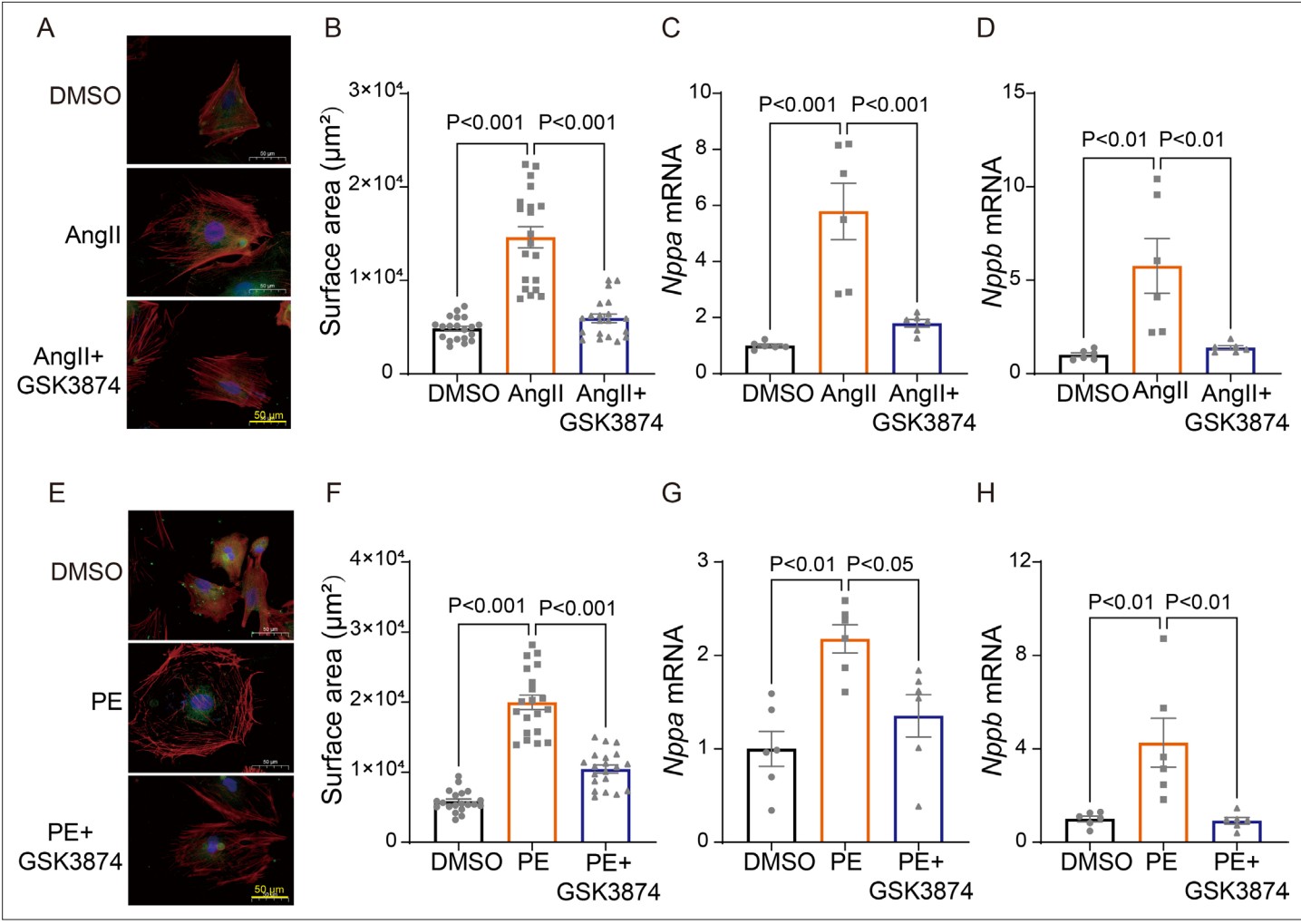

**Figure 6.** TRPV4 blockade attenuates AngII/PE-induced hypertrophy in NRVMs in vitro. Representative images (**A**) and statistics of the cell-surface areas (**B**) in NRVMs treated with DMSO, AngII, and AngII plus GSK3874 (n=20 cells from three animals). Statistics of ANP *(Nppa)* (**C**) and BNP *(Nppb)* (**D**) mRNA levels in NRVMs treated with DMSO, AngII, and AngII plus GSK3874 (n=6 per group). Representative images (**E**) and statistics of the cell-surface areas (**F**) in NRVMs treated with DMSO, PE, and PE plus GSK3874 (n=20 cells from three animals). Statistics of ANP *(Nppa)* (**G**) and BNP *(Nppb)* (**H**) mRNA levels in NRVMs treated with DMSO, PE, and PE plus GSK3874 (n=6 per group). All results represent mean ± SD, a one-way ANOVA followed by the Bonferroni test. NRVM, neonatal rat ventricular myocyte.

The online version of this article includes the following source data for figure 6:

**Source data 1.** Source data file (Excel) for *Figure 6B, C, D, F, G and H*.

GSK3874-treated mice (*Figure 5K*). In summary, these results indicate that GSK3874 treatment effectively reduces cardiac remodeling and dysfunction induced by TAC.

## The TRPV4 antagonist improves neonatal rat ventricular myocytes hypertrophy in vitro

Next, we sought to determine whether TRPV4 activation contributes to cardiomyocyte hypertrophy in vitro. Neonatal rat ventricular myocytes (NRVMs) were isolated from neonatal Sprague-Dawley (SD) rats and treated with angiotensin II (Ang II) or phenylephrine (PE) for 48 h. We found that AngII-stimulated cardiac hypertrophy, as indicated by increases in cell surface area (*Figure 6A–B*) and the expression of ANP *(Nppa)* and BNP *(Nppb)* (*Figure 6C–D*), was largely inhibited by the TRPV4 specific antagonist GSK3874. Similarly, PE-induced CM hypertrophy was also attenuated by GSK3874 (*Figure 6E–H*). Taken together, our results confirm that TRPV4 activation contributes to cardiac hypertrophy in vitro.

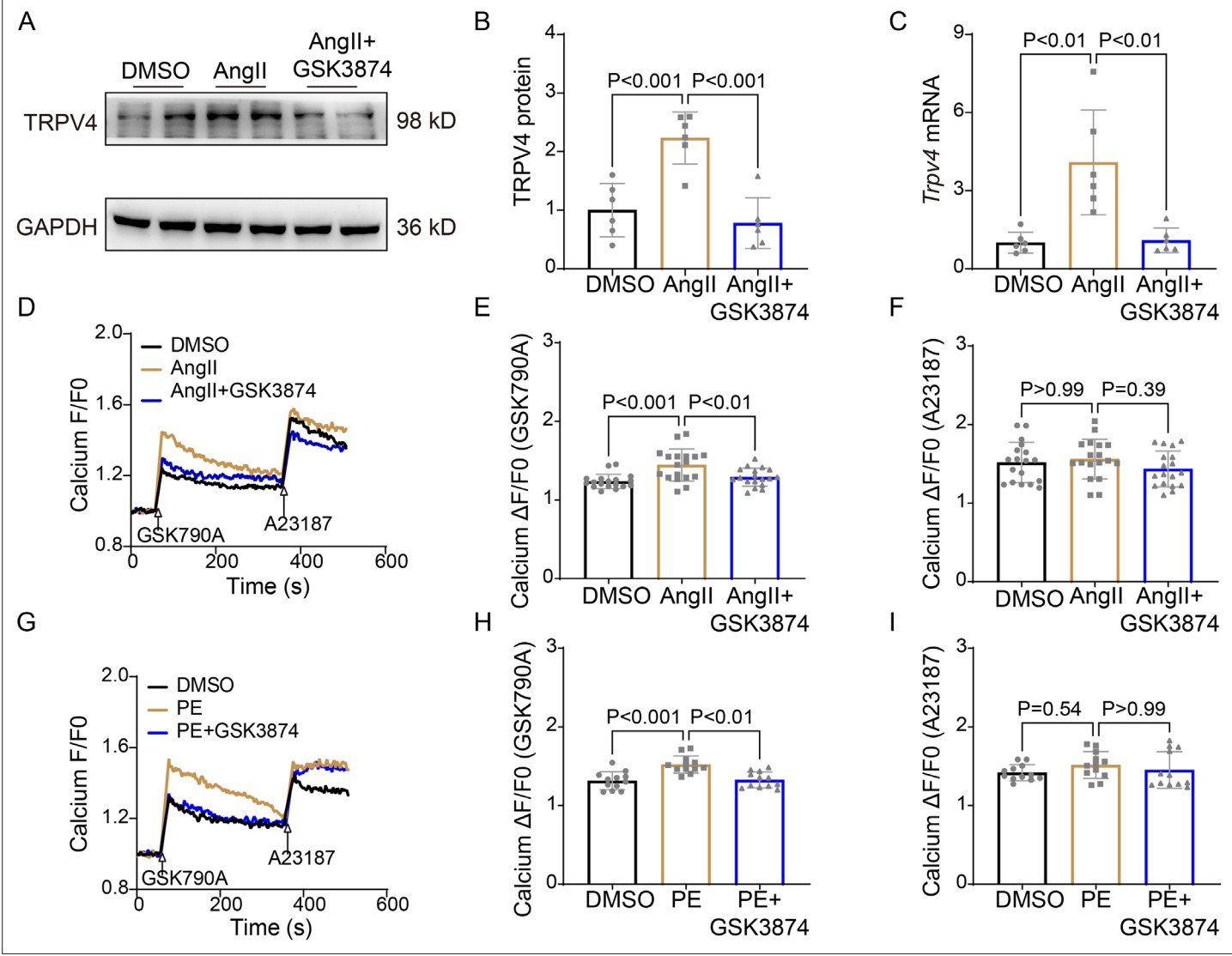

**Figure 7.** TRPV4 blockade attenuates AngII/PE-induced Ca$^{2+}$ overload in NRVMs. Representative immunoblot image (**A**) and statistics (**B**) of TRPV4 protein level in NRVMs treated with DMSO, AngII, and AngII plus GSK3874 (n=6 per group). (**C**). Statistical data of *Trpv4* mRNA level in NRVMs treated with DMSO, AngII, and AngII plus GSK3874 (n=6 per group). Representative recording of changes in intracellular Ca$^{2+}$ induced by 500 nM GSK 790A and 1 μM A23187 in NRVMs treated with DMSO, AngII, and AngII plus GSK3874 (**D**). Quantification of [Ca$^{2+}$]$_i$ response induced by GSK790A (**E**) and A23187 (**F**) -induced in NRVMs treated with DMSO, AngII, and AngII plus GSK3874 (n=18 per group). Representative recording of changes in intracellular Ca$^{2+}$ induced by 500 nM GSK 790A and 1 μM A23187 in NRVMs treated with DMSO, PE, and PE plus GSK3874 (**G**). Quantification of [Ca$^{2+}$]$_i$ response induced by GSK790A (**H**) and A23187 (**I**)-induced in NRVMs treated with DMSO, PE, and PE plus GSK3874 (n=12 per group). The arrow indicates the time of addition of GSK1016790A and A21387. F0 represents the average fluorescence intensity before GSK1016790A stimulation. All results represent mean ± SD, a one-way ANOVA followed by the Bonferroni test. NRVM, neonatal rat ventricular myocyte.

The online version of this article includes the following source data for figure 7:

**Source data 1.** Source data file (Excel) for *Figure 7B*.

**Source data 2.** Source data file (Excel) for *Figure 7C-I*.

## TRPV4 antagonist alleviates AngII/PE-induced Ca$^{2+}$ overload in NRVMs

It is well known that [Ca$^{2+}$]$_i$ increases in response to sustained hypertrophy. We have previously shown that TRPV4 is functionally expressed in cardiomyocytes and mediates Ca$^{2+}$ influx upon activation (*Wu et al., 2017b*). Here, we found that TRPV4 protein and mRNA expression were significantly increased in NRVMs after being treated with AngII (*Figure 7A–C* and *Source data 1*). To correlate TRPV4 expression to functional channel, changes in [Ca$^{2+}$]$_i$ in response to the specific TRPV4 agonist

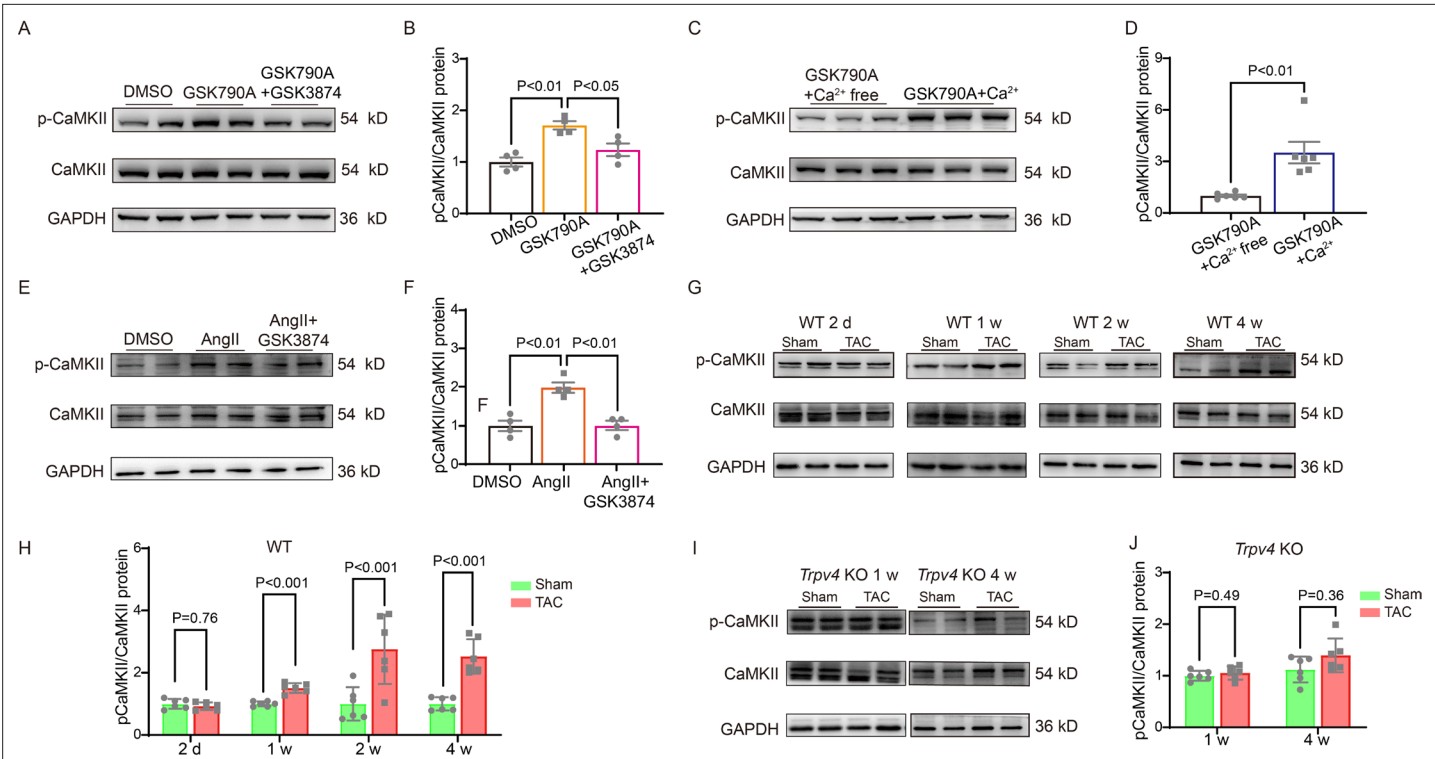

**Figure 8.** TRPV4 activation induces CaMKII phosphorylation. Representative immunoblot image (**A**) and statistics (**B**) of p-CaMKII/CaMKII in NRVMs treated with DMSO, GSK790A, and GSK790A plus GSK3874 (n=4 per group). All results represent mean ± SD, a one-way ANOVA followed by the Bonferroni test. Representative immunoblot image (**C**) and statistics (**D**) of p-CaMKII/CaMKII in NRVMs treated with GSK790A in the absence and presence of $Ca^{2+}$ medium (n=6 per group). All results represent mean ± SD, an unpaired two-tailed student's t-test. Representative immunoblot image (**E**) and statistics (**F**) of p-CaMKII/CaMKII in NRVMs treated with DMSO, AngII, and AngII plus GSK3874 (n=4 per group). All results represent mean ± SD, a one-way ANOVA followed by the Bonferroni test. Representative immunoblot image (**G**) and statistics (**H**) of p-CaMKII/CaMKII in the LV from sham or TAC WT mice at indicated time points after the operation (n=6 per group). All results represent mean ± SD, an unpaired two-tailed student's t-test. Representative immunoblot image (**I**) and statistics (**J**) of p-CaMKII/CaMKII in *Trpv4* KO mice 1 week or 4 weeks after sham or TAC operation (n=6 per group). All results represent mean ± SD, an unpaired two-tailed student's t-test. NRVM, neonatal rat ventricular myocyte.

The online version of this article includes the following source data for figure 8:

**Source data 1.** Source data file (Excel) for *Figure 8B, D, F, H and J*.

GSK1016790A (GSK790A, 500 nM), were measured in NRVMs after AngII stimulation. As shown in *Figure 7D–E*, GSK790A induced robust $Ca^{2+}$ influx, which was further enhanced after stimulation with AngII. However, pre-incubation of GSK3874 could inhibit this enhanced response. Note that treatment with AngII or AngII+GSK3874 did not affect $Ca^{2+}$ influx induced by A23187 (*Figure 7F*). Similar results were also obtained from NRVMs after PE stimulation (*Figure 7G–I*). Our results indicate that TRPV4 activation may be implicated in the $[Ca^{2+}]_i$ rise induced by sustained hypertrophy.

## TRPV4 activation contributes to CaMKII phosphorylated

$Ca^{2+}$/calmodulin-dependent protein kinase II (CaMKII) is upregulated after pressure overload and plays an essential role in cardiac hypertrophy and the progression of heart failure (*Ljubojevic-Holzer et al., 2020*; *Zhang et al., 2003*). More importantly, $Ca^{2+}$ entry via TRPV4 can activate CaMKII in many other cells (*Lyons et al., 2017*; *Woolums et al., 2020*; *Zhou et al., 2021*). Therefore, we hypothesized that TRPV4 activation contributes to cardiac hypertrophy through CaMKII. We first investigated the role of TRPV4 on CaMKII activation. Using NRVMs in vitro, we found that treatment with TRPV4 agonist GSK790A for 30 min markedly increased the expression of p-CaMKII (Thr287) compared with the DMSO group. However, GSK790A-induced activation of CaMKII was significantly blocked by either pretreating with TRPV4 antagonist GSK3874 or removing extracellular $Ca^{2+}$, demonstrating that TRPV4-mediated $Ca^{2+}$ influx promotes the activation of CaMKII (*Figure 8A–D* and *Source data 1*).

Consistent with previous studies (*Xiao et al., 2018*), NRVMs-stimulated AngII for 48 h showed a twofold increase in p-CaMKII, and this response was largely abrogated by pretreatment with GSK3874 (*Figure 8E–F*, *Source data 1*). We also examined the phosphorylation of CaMKII at 2 days, 1 week, 2 weeks, and 4 weeks after TAC in WT mice. Similar to our observation of TRPV4 expression, p-CaMKII began to increase 1 week after the operation and remained at a high level 4 weeks after TAC (*Figure 8G–H*, *Source data 1*). More interestingly, the TAC-induced upregulation of p-CaMKII in WT mice 1 or 4 weeks was not observed in *Trpv4* KO mice (*Figure 8I–J* and *Source data 1*). Our results indicate that TRPV4 activation is required for the phosphorylation of CaMKII in response to pressure overload.

## TRPV4 activation promotes NFκB phosphorylation via a CaMKII-dependent manner

As shown in *Figure 9A–B* and *Source data 1*, a short-term (30 min) treatment with TRPV4 agonist GSK790A also dramatically increased the level of phosphorylated NFκB p65 in NRVMs. This effect was abolished by pretreatment with TRPV4 antagonist GSK3874. Furthermore, AngII-induced phosphorylation of NFκB p65 was also prevented by pretreatment with GSK3874 (*Figure 9C–D* and *Source data 1*). Therefore, TRPV4 activation may promote the phosphorylation of NFκB p65.

We then asked how TRPV4 activation is linked to NFκB signaling. Since the phosphorylation of NFκB could be regulated by the CaMKII signaling pathway (*Ling et al., 2013*), we examined the involvement of CaMKII. Indeed, the application of the CaMKII inhibitor, KN93 (2 μM), but not its inactive analog KN92 (2 μM), abolished the GSK790A-stimulated NFκB p65 phosphorylation in NRVMs (*Figure 9E–F* and *Source data 1*), supporting the role of CaMKII in linking TRPV4-mediated Ca$^{2+}$ influx to NFκB activation.

## Discussion

In this study, we characterized the functional role of TRPV4 in pressure-induced cardiac hypertrophy and heart failure. We showed that TRPV4 activation promoted the development of pathological cardiac hypertrophy and heart failure. This effect was associated with Ca$^{2+}$-mediated CaMKII phosphorylation and subsequently the activation of NFκB-NLRP3 (*Figure 10*). These results suggest that TRPV4 may be a potential therapeutic target for cardiac hypertrophy and heart failure.

As a non-selective calcium ion channel, TRPV4 is widely expressed in the cardiovascular system and mediates cellular responses to a variety of environmental stimuli including hypo-osmolality, shear stress, and heat (*Hof et al., 2019*; *Randhawa and Jaggi, 2015*). Previous studies, including our own, have demonstrated that TRPV4 is functionally expressed in hearts (*Chaigne et al., 2021*; *Peana et al., 2022*; *Wu et al., 2017a*) and its mRNA or protein expression level can be upregulated by pressure overload (*Morine et al., 2016*), following ischemia-reperfusion (*Dong et al., 2017*; *Jones et al., 2019*; *Wu et al., 2017a*), under inflammation conditions (*Kumar et al., 2020*; *Liao et al., 2020*), as well as after the application of TRPV4 agonist GSK790A (*Adapala et al., 2016*). TRPV4 activation induces Ca$^{2+}$ influx and increases [Ca$^{2+}$]$_i$, which may subsequently promote cardiac remodeling and cardiac dysfunction. In the present study, we found that TRPV4 expression was significantly increased in mice hypertrophy hearts, human failing hearts, and AngII-induced hypertrophic cardiomyocytes, suggesting that TRPV4 was implicated in the processes of cardiac hypertrophy and failure. Previous studies have reported that Ang II can enhance TRPV4 activity and increase TRPV4 expression, through the activation of the AT1 receptor involving the PKC and Src kinase pathway, respectively (*Mercado et al., 2014*; *Saxena et al., 2014*). Future studies addressing TRPV4 upregulation in the context of cardiac hypertrophy are needed. Furthermore, the inhibition of TRPV4 by genetic deletion or antagonist GSK3874 attenuated TAC-induced cardiac hypertrophy and subsequence heart failure in vivo. Our in vitro experiments showed that TRPV4 blockade protected cardiac hypertrophy induced by AngII. Concomitant with this protection was the downregulation of multiple proteins and transcriptional markers associated with the initiation and the progression of hypertrophy, inflammation, fibrosis, and heart failure. However, the mechanosensitive of TRPV4 remains controversial. TRPV4 has been found to be directly activated by membrane stretch in *Xenopus laevis* oocytes (*Loukin et al., 2010*), but membrane stretch fails to open mammalian TRPV4 (*Nikolaev et al., 2019*). Therefore, TRPV4 could be a downstream signal initiated by a primary mechanoreceptor, such as Piezo1. Following membrane

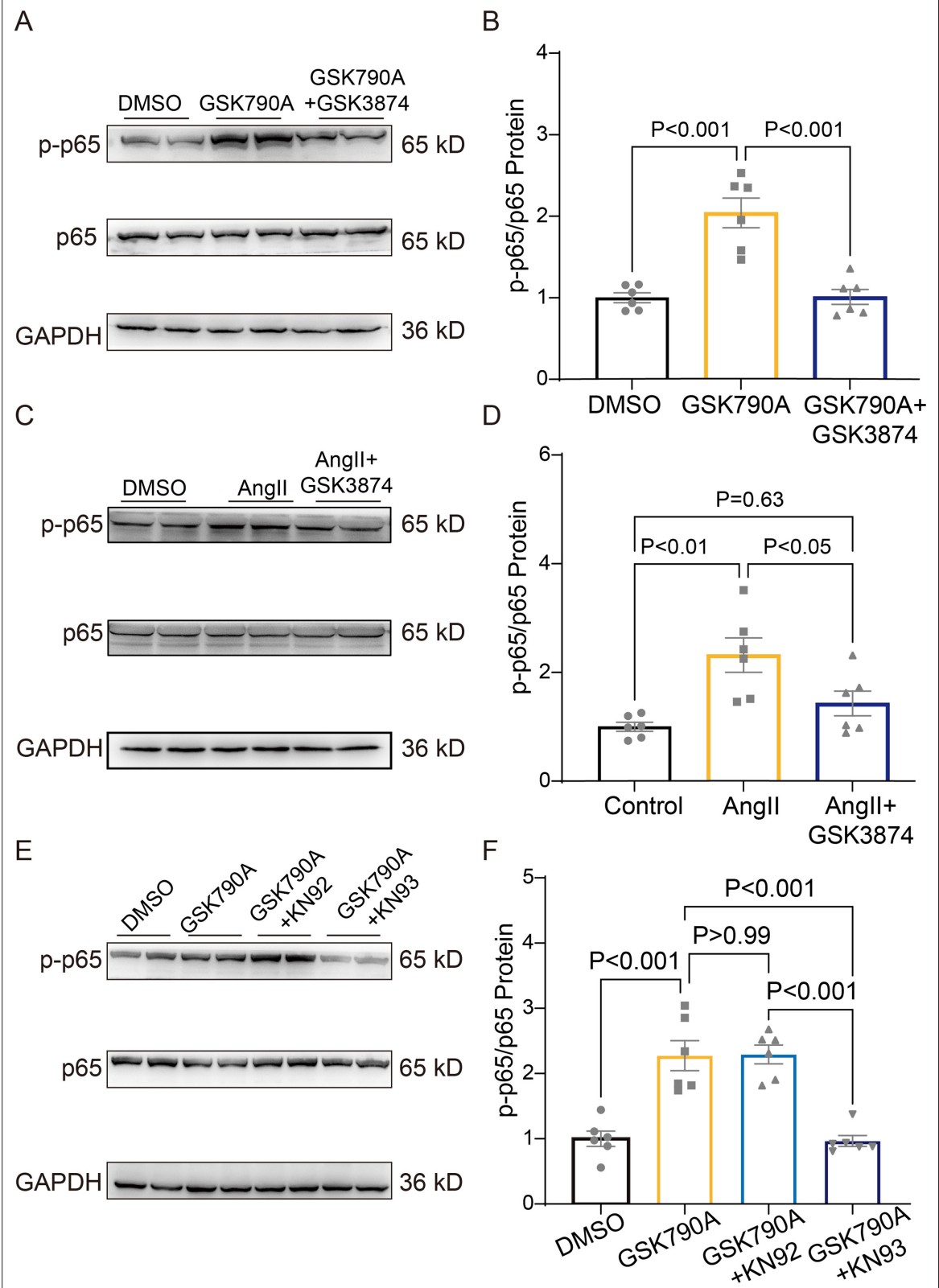

**Figure 9.** TRPV4 activation induces NF $\kappa$ B phosphorylation via a CaMKII signaling pathway. Representative immunoblot image (**A**) and statistics (**B**) of p-p65 /p65 in NRVMs treated with DMSO, GSK790A, and GSK790A plus GSK3874 (n=6 per group). All results represent mean ± SD, a one-way ANOVA followed by the Bonferroni test. Representative immunoblot image (**C**) and statistics (**D**) of p-p65 /p65 in NRVMs treated DMSO, AngII, and AngII plus GSK3874 (n=6 per group). All results represent mean ± SD, a one-way ANOVA followed by the Bonferroni test. Representative immunoblot image

*Figure 9 continued on next page*

*Figure 9 continued*

(**E**) and statistics (**F**) of p-p65 /p65 in NRVMs treated with DMSO, GSK790A, GSK790A plus KN92, and GSK790A plus KN93 (n=6 per group). All results represent mean ± SD, a one-way ANOVA followed by the Bonferroni test.

The online version of this article includes the following source data for figure 9:

**Source data 1.** Source data file (Excel) for *Figure 9B, D and F*.

stretch, Piezo1 is activated, causing a transient $Ca^{2+}$ influx as well as increased phospholipase A2 (PLA2) activity. PLA2-dependent arachidonic acid metabolites then activate TRPV4, which is observed after the osmotic stimulus and shear stress (*Gorelick and Nathanson, 2020*; *Watanabe et al., 2003*). Indeed, very recent studies have provided evidence that Piezo1 acts as the primary mechanoreceptor and initiates the cardiac hypertrophic response to pressure overload (*Guo et al., 2021a*; *Yu et al., 2021*). However, further studies are required to investigate the close interaction of TRPV4 with Piezo1 in pathological cardiac hypertrophy.

Recent studies have suggested that $Ca^{2+}$ influx through TRPV4 can result in the activation of CaMKII (*Lyons et al., 2017*; *Woolums et al., 2020*). CaMKII can be rapidly activated in response to pressure overload and plays an essential role in cardiac hypertrophy and decompensation to heart failure (*Baier et al., 2020*; *Swaminathan et al., 2012*). Therefore, we hypothesized that TRPV4 experiences mechanical stress, mediates $Ca^{2+}$ entry, and subsequently activates pro-hypertrophic signaling responses. Similar to our previous findings (*Wu et al., 2017a*), the TRPV4 agonist GSK790A induced robust $Ca^{2+}$ entry in NRVMs. We also found that GSK790A induced rapid phosphorylation of CaMKII, which could be prevented by TRPV4 antagonist and extracellular $Ca^{2+}$ removal, demonstrating that $Ca^{2+}$ entry following TRPV4 activation leads to CaMKII phosphorylation. Furthermore, AngII/PE-induced $[Ca^{2+}]_i$ rise, as well as the phosphorylation of CaMKII in NRVMs, was significantly reduced by the TRPV4 antagonist. Previous studies have also shown that Ang II/PE promotes the phosphorylation of CaMKII, increases CaMKII activity, and subsequently mediates cardiac hypertrophy (*Helmstadter et al., 2021*; *Nakamura and Sadoshima, 2018*; *Tonegawa et al., 2017*). However, the calcineurin/NFAT pathway may play a more important role in AngII-induced cardiac hypertrophy (*Yu et al., 2021*). In addition, our in vivo studies showed that the phosphorylation of CaMKII began to increase at 1 week and maintained higher levels 4 weeks after TAC, which was following the same trend of TRPV4 upregulation. More importantly, TAC-induced CaMKII phosphorylation was markedly blunted by genetic TRPV4 deletion. The evidence supports that TRPV4 plays a key role in mediating CaMKII activation during cardiac hypertrophy development.

Recent studies have shown that the activation of CaMKII triggers NFκB-NLRP3 activation and leads to inflammation, which is important for the initiation and progression of pathological cardiac hypertrophy (*Suetomi et al., 2018*; *Willeford et al., 2018*). We found that TAC induced increases in IL-1β, IL-6, TNF-α, MIP-2, and MCP-1 expression. Meanwhile, the phosphorylation of p-65 and the expression of NLRP3, ASC, and cleaved caspase-1 were upregulated in WT mice. The above-enhanced effects, however, were diminished in *Trpv4* KO mice. Similarly, AngII/PE-induced upregulation of p-65 phosphorylation in NRVMs was reduced by pretreatment with a TRPV4 antagonist. These results suggest that TRPV4 activation promotes NFκB-NLRP3 activation and inflammation in response to pressure overload, which further demonstrates a mechanism for TRPV4 in this response. Several other studies have also found that TRPV4 activation induces inflammation through the NFκB-NLRP3 signaling pathway (*Wang et al., 2021*; *Wang et al., 2019b*). In addition, we found that GSK790A also induced rapid phosphorylation of NFκB, which could be prevented by KN-93 for CaMKII inhibition, and this implies that CaMKII is involved in TRPV4 activation-induced phosphorylation of NFκB. Therefore, our data continue to highlight the importance of TRPV4-mediated $Ca^{2+}$ in intracellular signaling pathways and raise the possibility that TRPV4 activation promotes $Ca^{2+}$ influx, leads to the phosphorylation of CaMKII, and subsequently triggers the activation of NFκB-NLRP3, thus contributing to adverse cardiac remodeling.

An important limitation of our investigation is the use of the systemic functional abrogation TRPV4 model. TRPV4 is also expressed in cardiac fibroblasts and endothelial cells. Therefore, the effect of TRPV4 deletion on cardiac remodeling and dysfunction is not limited to cardiomyocytes. Indeed, the deletion of endothelial TRPV4 has been found to suppress TAC-induced cardiac hypertrophy and dysfunction via increased coronary angiogenesis and reduced cardiac fibrosis (*Adapala et al., 2019*). Interactions with cardiac fibroblasts or endothelial cells will need further study. Although the

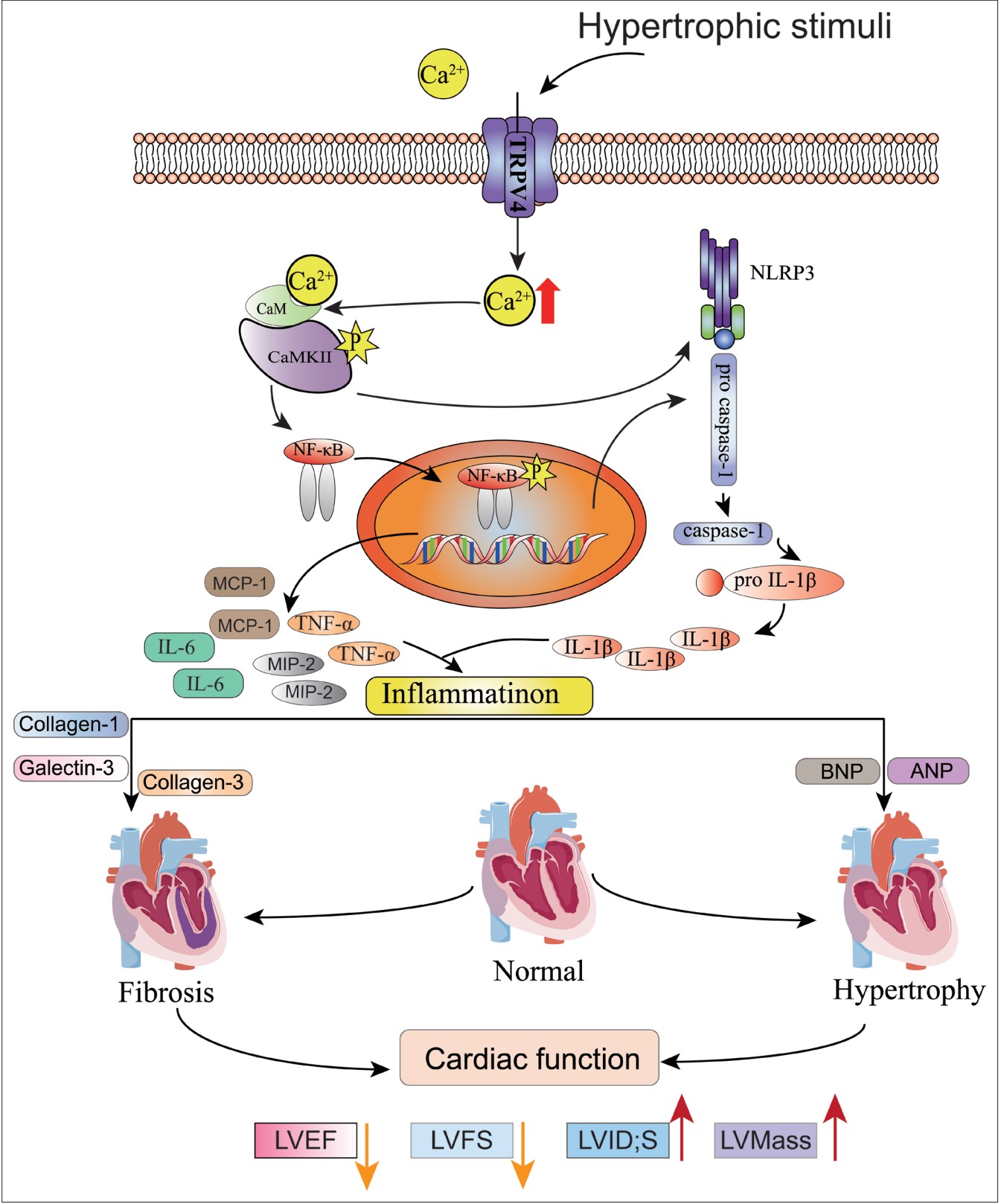

**Figure 10.** Schematic illustration of potential mechanisms through which TRPV4 activation promotes pathological cardiac hypertrophy.

upregulation of TRPV4 was consistent in mouse hypertrophy hearts and human failing hearts, our data do not provide conclusive evidence about the involvement of TRPV4 in hypertensive cardiac damage in patients. Further human studies are needed to verify our results.

Collectively, our findings imply that TRPV4 might be a stress response molecule that is upregulated in cardiac hypertrophy. Activation of TRPV4 induced increases in $Ca^{2+}$ influx, activated CaMKII, enhanced pro-inflammatory NFκB-NLRP3 signaling, and promoted inflammation response, thus contributing to pathological cardiac remodeling. TRPV4 antagonism provides an exploitable therapeutic advantage for the treatment of cardiac hypertrophy and subsequent heart failure.

# Materials and methods

**Key resources table**

| Reagent type (species) or resource | Designation | Source or reference | Identifiers | Additional information |
|---|---|---|---|---|
| Chemical compound, drug | Ang II | MCE | Cat# HY-13948 | |
| Chemical compound, drug | PE | MACKLIN | Cat# I822933 | |
| Chemical compound, drug | GSK790A | Sigma-Aldrich | Cat# G0798 | |
| Chemical compound, drug | GSK3874 | Sigma-Aldrich | Cat# SML0942 | |
| Chemical compound, drug | A21387 | Sigma-Aldrich | Cat# G0798 | |
| Chemical compound, drug | KN92 | Selleck | Cat# S6507 | |
| Chemical compound, drug | KN93 | Selleck | Cat# S6787 | |
| Chemical compound, drug | Pentobarbital sodium | Sigma-Aldrich | Cat# 76-74-4 | |
| Chemical compound, drug | Collagenase II | Worthington | Cat# ls004176 | |
| Chemical compound, drug | Fluo-4/AM | AAT Bioquest | Cat# AAT-B20401 | |
| Chemical compound, drug | Pluronic F-127 | Solarbio | Cat# P679 | |
| Sequence-based reagent | BNP (*Nppb*)_R (mice) | This paper | PCR primers | CAACTTCAGTGCGTTACAGC |
| Sequence-based reagent | Collagenase-1 (*Col1a1*)_F (mice) | This paper | PCR primers | GAAACCCGAGGTATGCTTGA |
| Sequence-based reagent | Collagenase1 (*Col1a1*)_R (mice) | This paper | PCR primers | GGGTCCCTCGACTCCTACAT |
| Sequence-based reagent | Collagenase-3 (*Col3a1*)_F (mice) | This paper | PCR primers | AGCCACCTTGGTCAGTCCTA |
| Sequence-based reagent | Collagenase-3 (*Col3a1*)_R (mice) | This paper | PCR primers | GTGTAGAAGGCTGTGGGCAT |
| Sequence-based reagent | Galectin-3 (*LGALS3*)_F (mice) | This paper | PCR primers | CAGGAAAATGGCAGACAGCTT |
| Sequence-based reagent | Galectin-3 (*LGALS3*)_R(mice) | This paper | PCR primers | CCCATGCACCCGGATATC |
| Sequence-based reagent | IL-1β (*Il1b*)_F (mice) | This paper | PCR primers | TGCCACCTTTTGACAGTGATG |
| Sequence-based reagent | IL-1β(*Il1b*)_R (mice) | This paper | PCR primers | TGATGTGCTGCTGCGAGATT |
| Sequence-based reagent | IL-6(*Il6*)_F (mice) | This paper | PCR primers | GATAAGCTGGAGTCACAGAAGG |
| Sequence-based reagent | IL-6(*Il6*)_R (mice) | This paper | PCR primers | TTGCCGAGTAGATCTCAAAGT |
| Sequence-based reagent | TNF-α(*Tnfa*)_F (mice) | This paper | PCR primers | CCCCAAAGGGATGAGAAGTT |
| Sequence-based reagent | TNF-α(*Tnfa*)_R (mice) | This paper | PCR primers | ACTTGGTGGTTTGCTACGA |
| Sequence-based reagent | MIP-2(*Mip2*)_F (mice) | This paper | PCR primers | CGCCCAGACAGAAGTCATAG |
| Sequence-based reagent | MIP-2(*Mip2*)_R (mice) | This paper | PCR primers | TCCTCCTTTCCAGGTCAGTTA |

*Continued on next page*

*Continued*

| Reagent type (species) or resource | Designation | Source or reference | Identifiers | Additional information |
|---|---|---|---|---|
| Sequence-based reagent | MCP-1(*Mcp1p*)_F (mice) | This paper | PCR primers | TTTTTGTCACCAAGCTCAAGAG |
| Sequence-based reagent | MCP-1(*Mcp1p*)_R (mice) | This paper | PCR primers | TTCTGATCCTCATTTGGTTCCGA |
| Sequence-based reagent | *Gapdh*_F(mice) | This paper | PCR primers | AAGAAGGTGGTGAAGCAGGCAT |
| Sequence-based reagent | *Gapdh*_F (mice) | This paper | PCR primers | CGGCATCGAAGGTGGAAGAGTG |
| Sequence-based reagent | *Trpv4*_F (rat) | This paper | PCR primers | CGTCCAAACCTGCGAATGAAGTTC |
| Sequence-based reagent | *Trpv4*_F (rat) | This paper | PCR primers | CCTCCATCTCTTGTTGTCACTGG |
| Sequence-based reagent | ANP (*Nppa*)_F (rat) | This paper | PCR primers | ATCTGATGGATTTCAAGAACC |
| Sequence-based reagent | ANP (*Nppa*)_R (rat) | This paper | PCR primers | CTCTGAGACGGGGTTGACTTC |
| Sequence-based reagent | BNP (*Nppb*)_F (rat) | This paper | PCR primers | CAATCCACGATGCAGAAGCT |
| Sequence-based reagent | BNP (*Nppb*)_R (rat) | This paper | PCR primers | GGGCCTTGGTCCTTTGAGA |
| Sequence-based reagent | *Gapdh*_ F (rat) | This paper | PCR primers | ATGGGAAGCTGGTCATCAAC |
| Sequence-based reagent | *Gapdh*_ R (rat) | This paper | PCR primers | GTGGTTCACACCCATCACAA |
| Antibody | Anti-GAPDH HRP (mouse monoclonal) | Bioworlde | Cat# MB001H RRID: AB_2857326 | Western blot (1:10,000) |
| Antibody | Anti-TRPV4 (rabbit polyclonal) | Alomone labs | Cat# ACC-034 RRID: AB_2040264 | Western blot (1:500) |
| Antibody | Anti-p-CaMKII(Thr287) (rabbit polyclonal) | Thermo Fisher Scientific | Cat# PA5-37833 RRID: AB_2554441 | Western blot (1:500) |
| Antibody | Anti-CaMKII (rabbit monoclonal) | Abcam | Cat# ab52476 RRID: AB_868641 | Western blot (1:1000) |
| Antibody | Anti-p-P65 (Ser536) (rabbit polyclonal) | Affinity | Cat# AF2006 RRID: AB_2834435 | Western blot (1:500) |
| Antibody | Anti-P65 (rabbit polyclonal) | Affinity | Cat# AF5006 RRID: AB_2834847 | Western blot (1:500) |
| Antibody | Anti-NLRP3 (rabbit monoclonal) | Abcam | Cat# ab263899 RRID: AB_2889890 | Western blot (1:500) |
| Antibody | Anti-ASC/TMS1 (rabbit monoclonal) | CST | Cat# 67824 RRID: AB_2799736 | Western blot (1:1000) |
| Antibody | Anti-Cleaved-Caspase 1, p20 (rabbit polyclonal) | Affinity | Cat# AF4005 RRID: AB_2845463 | Western blot (1:500) |
| Antibody | Anti-IL 1β (rabbit monoclonal) | Abcam | Cat# ab234437 | Western blot (1:1000) |
| Antibody | Anti-IL-6 (rabbit monoclonal) | CST | Cat# 12912 RRID: AB_2798059 | Western blot (1:500) |
| Antibody | Anti-TNF-α (rabbit monoclonal) | CST | Cat# 11948 RRID: AB_2687962 | Western blot (1:1000) |
| Antibody | Goat Anti-Rabbit IgG HRP (goat polyclonal) | Affinity | Cat# S0001 RRID: AB_2839429 | Western blot (1:3000) |
| Antibody | Anti-α-actinin (rabbit polyclonal) | Abcam | Cat# Ab137346 RRID: AB_2909405 | ICC (1:500) |
| Commercial assay or kit | cDNA reverse transcription kit | Vazyme | Cat# R211-01 | |

*Continued on next page*

*Continued*

| Reagent type (species) or resource | Designation | Source or reference | Identifiers | Additional information |
|---|---|---|---|---|
| Commercial assay or kit | SYBR Green PCR Master Mix Kit | CWbio | Cat# cw3008h | |
| Others | Percoll | Cytiva | Cat# 17089109 | A low-viscosity, non-toxic medium suitable for density gradient centrifugation of cells, viruses and subcellular particles |

## Human heart tissues

Explanted, heart failure tissues were obtained from five patients with dilated cardiomyopathy (DCM) undergoing cardiac transplantation. All patients were diagnosed with DCM with EF less than 40% at least 3 months prior to heart transplantation. Non-heart failure tissues were obtained from three organ donors whose hearts could not be placed due to size issues, ABO mismatch, or other factors. The clinical data of patients have been shown in *Appendix 1—table 2*. The study was in accordance with the Declaration of Helsinki (as revised in 2013). The study was reviewed and approved by the Ethics Committee of Union Hospital, Tongji Medical College, Huazhong University of Science and Technology (Wuhan, China; approval number: UHCT-IEC-SOP-016-03-01). Written informed consent and consent to publish were obtained from all the patients.

## Animals

Male C57BL/6 mice and newborn SD rats were purchased from the Laboratory Animal Center, Xuzhou Medical University (Xuzhou, China). *Trpv4* KO mice were generated on the C57BL/6 background, as described previously (*Dong et al., 2017*; *Mizuno et al., 2003*). Genotyping was performed by PCR using ear punch/tail snip biopsies with the following primers: WT forward primer 5′-TGTTC GGGTGGTTTGGCCAGGATAT-3′ and reverse primer 5′-GGTGAACCAAAGGACACTTGCATAG-3′, which produce a 796 bp product from the WT allele; knockout forward primer 5′-GCTGCATACGCTT GATCCGGCTAC-3′ and reverse primer 5′-TAAAGCACGAGG AAGCGGTCAGCC-3′, which produce a 366-bp product from the target allele (*Appendix 1—figure 1A*). RT-PCR of heart mRNA was used to confirm the deletion of TRPV4 sequence, indicated by a 534-bp cDNA fragment of WT mice, but absent in *Trpv4* KO mice (*Appendix 1—figure 1B*), as previously described (*Boudaka et al., 2020*). All animal protocols were performed in adherence to the National Institutes of Health Guidelines and were approved by the Experimental Animal Ethics Committee of Xuzhou Medical University (Xuzhou, China; approval number: 202204A084). Animals were housed in a temperature-regulated room (12 h day/12 h night cycle) with ad libitum access to food and water.

## TAC surgery and treatment

Eight- to twelve-week-old male WT and *Trpv4* KO mice were subjected to TAC to induce pressure overload. Mice were anesthetized by intraperitoneal (i.p.) injection of pentobarbital sodium (50 mg/kg), intubated via the oral cavity, and ventilated at 110 breaths/min. Following a sternotomy, the transverse aorta between the right innominate and left carotid arteries was dissected and banded with a blunt L type 27-gauge needle using a 5-0 silk suture. The needle was then removed. Successful TAC surgery was confirmed by measuring the right carotid/left carotid flow velocity ratio. The sham-operated mice underwent an identical procedure but without aortic constriction. WT mice were treated with vehicle (6% Cavitron) or GSK3874 (10 mg/kg/d) via oral gavage for 4 weeks after TAC (*Liao et al., 2020*; *Thorneloe et al., 2012*).

## Echocardiography

Echocardiography was performed by using a Vevo 2100 Ultrasound System (Visual Sonics, Toronto, Canada), as described in a previous study (*Chen et al., 2019*). Briefly, the mice were anesthetized with isoflurane. Parasternal long- and short-axis views in B- and M-Mode were recorded when the heart rate of the mice was maintained at 430–480 beats/min. The EF, FS, left ventricular end-systolic diameter (LVID), LV mass, and other function parameters were calculated with Vevo LAB software (Visual Sonics) by a technician who was blinded to the treatment groups.

## Tissue collection

After the echo examination, the heart was harvested and rinsed with cold phosphate-buffered saline. After being weighted, the LV was cut into two parts. The top part was put into 4% paraformaldehyde for histological analysis, and the bottom part was quickly put into liquid nitrogen and transferred to a –80° freezer later. The HW normalized to BW and TL were measured as indicators of cardiac hypertrophy (*Zhao et al., 2016*).

## Histological analyses

For histological analysis, transverse LV sections were cut into 4 µm slices. The hematoxylin and eosin staining was performed to analyze the histological change. Masson's trichrome stain was performed to assess cardiac fibrosis. FITC-conjugated wheat germ agglutinin was performed to determine cell size. A quantitative digital image analysis system (ImageJ software) was used in image measurement.

## Isolation of NRVMs and treatment

NRVMs were isolated according to previously established protocols (*Golden et al., 2012*). In brief, LV from 1- to 3-day-old SD rats was harvested and digested in the presence of 0.5 mg/ml collagenase II at 37°C. NRVMs were further purified by Percoll gradient centrifugation. Cells were plated at a density of $2.5 \times 10^5$ cells/cm$^2$ on collagen-coated plates and cultured in Dulbecco's modified Eagle's medium (DMEM) supplemented with 15% fetal bovine serum (FBS; Hyclone, USA), 100 units/ml penicillin, 100 µg/ml streptomycin, and 2 µg/ml cytosine arabinoside. The next morning, the media were changed to FBS-free DMEM for 24 hr. Ventricular myocyte hypertrophy was induced by treatment with Ang II or PE for 48 hr. In another group of experiments, cells were treated with TRPV4 agonist GSK790A (500 nM) according to the time required for the experiment, while TRPV4 antagonist GSK3874 (300 nM), KN92 (2.0 µM), and KN93 (2.0 µM) was applied 30 min earlier.

## Assessment of cell surface area

NRVMs were stained with antibodies for sarcomeric α-actinin and cell nuclei were counterstained with DAPI. Cell size was examined by TRITC-phalloidin staining assay and measured with ImageJ software.

## Calcium fluorescence

Calcium imaging was performed as previously described (*Wang et al., 2019a*; *Wu et al., 2017b*). NRVMs were loaded with Fluo-4/AM (2 µM) and F-127 (0.03%) for 30 min. Cells in 96-wells plates were illuminated at 488 nm and fluorescence emissions at 525 nm were captured by a multifunctional microplate reader (TECAN, Infinite 200PRO, Swiss). Cells were stimulated with the TRPV4 agonist GSK790A (500 nM). A21387 (1 µM) was set as a positive control.

## RNA extraction, cDNA synthesis, and quantitative PCR

Total RNA was extracted from LV tissues or cultured NRVMs using the Extraction Kit according to the manufacturer's instructions. For cDNA synthesis, 500 ng RNA was reverse transcribed using a high-capacity cDNA reverse transcription kit. Real-time quantitative PCR (qPCR) was performed with SYBR Green PCR Master Mix Kit on a QuantStudio 3 system (Applied Biosystems, Foster City, CA). GAPDH was used as a housekeeper gene for the normalization of gene expression. The primers used in qPCR were listed in the Key resources table. The result for each gene was obtained from at least six independent experiments.

## Western blots

Total protein was extracted from LV tissues or cultured NRVMs with RIPA reagent. Then, protein expression was analyzed by standard western blot as described previously (*Wu et al., 2020*). Briefly, protein (30 µg for each sample) was separated using 10% SDS-polyacrylamide gel electrophoresis and subsequently transferred onto polyvinylidene difluoride membranes (Millipore, Darmstadt, Germany). After 1 hr of blocking with Western blocking buffer (CWbio, Taizhou, China), the membranes were incubated with primary antibody at 4°C. The next day, the membranes were washed with TBST and incubated with corresponding horseradish peroxidase (HRP)-conjugated secondary antibodies for 1 hr at room temperature. Finally, proteins were visualized with the enhanced chemiluminescence kit (Affinity, Ancaster, ON, Canada). Band intensity was quantified by Tanon image plus software (Tanon,

Nanjing, China). GAPDH was used as a loading control. The antibodies used in the study were listed in the Key resources table.

## Statistical analysis

All statistical data were presented as mean ± SD and analyzed by Graphpad prism 9. An unpaired two-tailed Student's t-test was used for comparison between the two groups. The differences among multiple groups were analyzed using one-way ANOVA or two-way ANOVA followed by the Bonferroni adjustment for multiple comparisons. $p < 0.05$ was reported as statistically significant.

## Acknowledgements

The authors thank Prof. Atsuko Mizuno (Jichi Medical University, Japan) for *Trpv4* KO mice. Dr. Yimei Du thanks Jenny Xiao (Columbia University, New York, USA) for editing the manuscript and checking for grammatical errors. This study was supported by the National Natural Science Foundation of China (82170326 and 81770328 to Y.D.) and by the National Health Commission of Xuzhou (XWKYHT20200069).

## Additional information

### Funding

| Funder | Grant reference number | Author |
| --- | --- | --- |
| National Natural Science Foundation of China | 82170326 | Yimei Du |
| National Natural Science Foundation of China | 81770328 | Yimei Du |

The funders had no role in study design, data collection and interpretation, or the decision to submit the work for publication.

### Author contributions

Yan Zou, Data curation, Formal analysis, Investigation, Validation, Writing – original draft, Writing – review and editing; Miaomiao Zhang, Data curation, Formal analysis, Investigation, Validation; Qiongfeng Wu, Resources; Ning Zhao, Software; Minwei Chen, Methodology, Validation; Cui Yang, Methodology, Visualization; Yimei Du, Conceptualization, Funding acquisition, Writing – original draft, Writing – review and editing; Bing Han, Funding acquisition, Project administration, Supervision, Writing – review and editing

### Author ORCIDs

Yimei Du ⬚ http://orcid.org/0000-0003-1125-0294

### Ethics

Human subjects: The study was reviewed and approved by the Ethics Committee of Union Hospital, Tongji Medical College, Huazhong University of Science and Technology (Wuhan, China; approval number: UHCT-IEC-SOP-016-03-01). Written informed consent and consent to publish was obtained from all the patients.

All animal protocols were performed in adherence to the National Institutes of Health Guidelines and were approved by the Experimental Animal Ethics Committee of Xuzhou Medical University (Xuzhou, China; approval number:202204A084).

### Decision letter and Author response

Decision letter https://doi.org/10.7554/eLife.74519.sa1
Author response https://doi.org/10.7554/eLife.74519.sa2

## Additional files

### Supplementary files
• Appendix 1—table 1—source data 1.
• Appendix 1—figure 1—source data 1.
• Transparent reporting form
• Source data 1. Raw Western blot image.

### Data availability
All data generated or analyzed during this study are included in the manuscript and supporting file. Source Data files have been provided for Figures 1-9.

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

# Appendix 1

**Appendix 1—table 1.** Anatomical parameters, early markers of cardiac hypertrophy and fibrosis, and echocardiographic parameters were measured in wild-type (WT) mice after subjected to TAC versus sham-operated controls (n=6–7/group).

| | 2 d | | 1 w | | 2 w | | 4 w | |
|---|---|---|---|---|---|---|---|---|
| | Sham | TAC | Sham | TAC | Sham | TAC | Sham | TAC |
| Anatomical parameter | | | | | | | | |
| n | 7 | 7 | 7 | 7 | 7 | 7 | 6 | 6 |
| HW/BW | 4.4±0.1 | 4.5±0.1 | 5.1±0.2 | 7.3±0.7*** | 4.5±0.3 | 7.1±1.1*** | 4.5±0.3 | 7.1±0.8** |
| HW/TL | 6.0±0.2 | 6.0±0.1 | 6.7±0.6 | 8.4±0.6*** | 6.0±0.3 | 9.3±1.4*** | 6.3±0.5 | 9.7±1.6*** |
| Early markers of cardiac hypertrophy and fibrosis | | | | | | | | |
| n | 6 | 6 | 6 | 6 | 6 | 6 | 7 | 7 |
| ANP | 1.0±0.3 | 1.1±0.2 | 1.0±0.5 | 9.1±3.9*** | 1.0±0.4 | 28.1±11.5*** | 1.0±0.2 | 7.3±1.9*** |
| BNP | 1.0±0.2 | 0.7±0.1 | 1.0±0.4 | 11.5±2.0*** | 1.0±0.3 | 5.2±1.1*** | 1.0±0.4 | 2.7±0.4** |
| Collagenase-1 | 1.0±0.4 | 1.4±0.7 | 1.0±0.4 | 1.1±0.3 | 1.0±1.3 | 1.0±0.7 | 1.0±0.2 | 6.8±2.5** |
| Collagenase-3 | 1.0±0.2 | 1.0±0.5 | 1.0±0.4 | 1.30±0.25 | 1.0±1.1 | 2.5±2.1 | 1.0±0.3 | 3.8±1.2** |
| Echocardiographic parameters | | | | | | | | |
| n | 7 | 7 | 7 | 7 | 7 | 7 | 6 | 6 |
| EF (%) | 66.4±7.2 | 65.0±8.9 | 71.0±5.9 | 68.7±3.5 | 75.7±4.4 | 60.2±3.9*** | 68.8±3.3 | 52.8±10.6** |
| FS (%) | 36.0±5.1 | 34.2±6.7 | 39.8±6.6 | 36.1±3.5 | 43.4±4.0 | 31.3±2.6*** | 37.6±2.7 | 26.9±6.5** |

HW/BW: heart weight to body weight ratio, HW/TL: heart weight to tibia length ratio, EF: ejection fraction, FS: fractional shortening, Data represent means ± SD, **$P<0.01$, ***$P<0.001$, compared between sham- and TAC operated groups.

The online version of this article includes the following source data for appendix 1—table 1:

• **Appendix 1—table 1—source data 1.**

**Appendix 1—table 2.** Clinical characteristics of patients with advanced heart failure.

| Patient | Age (y) | Gender | EF | NYHA |
|---|---|---|---|---|
| DCM 1 | 52 | Male | 22.5 | IV |
| DCM 2 | 47 | Male | 12.6 | III |
| DCM 3 | 46 | Male | 15.9 | IV |
| DCM 4 | 20 | Male | 15 | III |
| DCM 5 | 30 | Female | 21 | IV |
| Control 1 | 35 | Male | | I |
| Control 2 | 27 | Male | | I |
| Control 3 | 40 | Male | | I |

DCM: dilated cardiomyopathy, EF: ejection fraction, NYHA: New York Heart Association

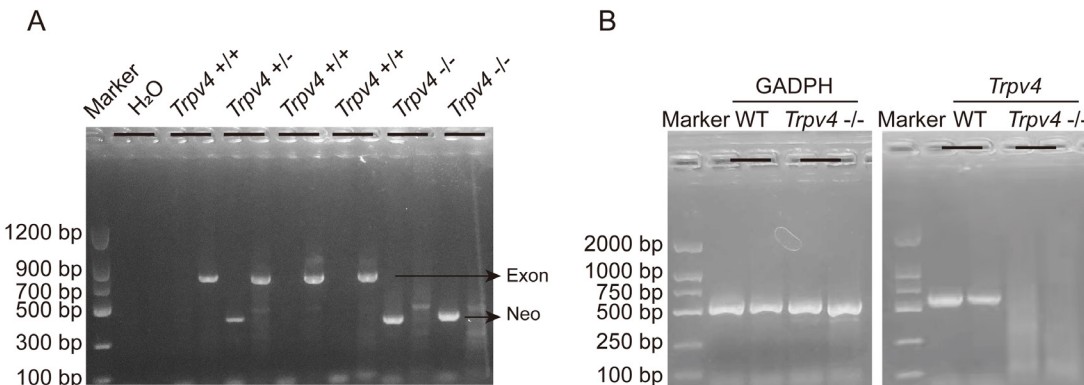

**Appendix 1—figure 1.** Genotyping of TRPV4 wild-type and Trpv4 KO mice and TRPV4 depletion in the heart of Trpv4 KO mice. A. Representative RT-PCR genotyping gel image of the WT, Trpv4+/-, and Trpv4-/-. B. RT-PCR of total RNA from heart showing Trpv4 mRNA was present in WT mice but absent in Trpv4 KO mice.

The online version of this article includes the following source data for appendix 1—figure 1:

• **Appendix 1—figure 1—source data 1.**

