## [Editor Report]

Zou et al., demonstrate that activation of the TRPV4 ion channel is a novel contributor to cardiac hypertrophy, adding to the number of mechanisms known to trigger pathological left ventricular hypertrophy (LVH), While the exact signaling pathway downstream of TRPV4 leading to pathological hypertrophy still remain to be uncovered, the findings that this ion channel contributes to cardiac hypertrophy and that the Piezo1 channel acts upstream of PLA2 and TRPV4 add to our understanding of the pathogenesis of LVH and heart failure.

---

## [Decision Letter]

**Decision letter after peer review:**

Thank you for submitting your article "Activation of transient receptor potential vanilloid 4 is involved in pressure overload-induced cardiac hypertrophy" for consideration by *eLife*. Your article has been reviewed by 3 peer reviewers, one of whom is a member of our Board of Reviewing Editors, and the evaluation has been overseen by a Senior Editor. The following individual involved in review of your submission has agreed to reveal their identity: Boris Martinac (Reviewer #3).

The reviewers have discussed their reviews with one another, and the Reviewing Editor has drafted this letter to help you prepare a revised submission.

Essential revisions:

Despite providing experimental evidence for involvement of the TRPV4 ion channels in LVH and cardiac failure, this study does not yet provide a clear mechanistic explanation about the role of TRPV4 in the cardiac pathology. Significantly more work is required to clarify this issue. In particular, the TRPV4 protein and mRNA expression should have been determined at earlier time points, i.e. at several days or two weeks rather than only at four weeks. Furthermore, as also pointed out by the authors, examining only the effects of a systemic functional deletion of TRPV4 was considered a limitation whereas a cardiomyocyte-specific deletion of TRPV4 in a mouse model would have strengthened the study.

1) To examine if TRPV4 plays a role in cardiac hypertrophy, the authors measured the TRPV4 protein and mRNA expression levels in LV tissue from WT and sham-operated BL/6 mice at a single time point of four weeks after TAC surgery. However, it has been demonstrated that after the period longer than two weeks the hearts of TAC operated BL/6 mice aged 10 or 12 weeks manifested cardiac dysfunction by progressing from hypertrophy to heart failure when additional signaling pathways become activated compared to LVH only. This is also indicated by the findings of this study showing "the progression of hypertrophy, inflammation, fibrosis, and heart failure" (p. 21, line 315). Thus, there is no clear distinction between the TRPV4 involvement in development of LVH vs heart failure. This is further addressed in point 3 below.

2) It is unclear which type of "mechanical loading" (p. 21, line 299) is activating TRPV4. Given the authors' suggestion that TRPV4 activation in pathological cardiac hypertrophy and heart failure occurs in response to pressure overload, does it mean that TRPV4 is activated by membrane stretch? However, this is unlikely because mammalian TRP ion channels are not activated by membrane stretch (Nikolaev et al., J Cell Sci 2019). Consequently, TRPV4 may not play a role as an initiating stressor but is more likely functioning as an upstream signaling transducer (p. 21, line 316). What is then acting upstream of TRPV4 as the primary transducer of the pressure overload? The current study does not answer this relevant question.

3) Yu et al. (Front. Cell Dev. Biol. 2021) reported two ca^2+^-calmodulin dependent hypertrophic pathways leading to LVH; the first one via AngII acting on Gq-coupled receptors and the second one via TAC-induced pressure overload. Both pathways are ca^2+^-calmodulin dependent with the AngII one via the ca^2+^-calmodulin dependent activation of calcineurin and the TAC one via ca^2+^ -calmodulin dependent activation of CaMKII. What is obscuring the results of this study is that the authors use AngII/PE model of cardiac dysfunction (e.g. p. 22, lines 327-32), which although exhibiting hypertrophy, is unrelated to the TAC-induced LVH. Consequently, the signaling pathways are different, and it remains unclear how TRPV4 is activated in this case. Further complication with the phenylephrine is that in addition to acting as a α1-adrenoceptor agonist it also causes ca^2+^ store depletion leading to ca^2+^-induced calcium release from SR through TRPV4 channels (Dryn et al., Int. J. Physiol. Pathophysiol. 2017).

4) The data of the TAC model are convincing, and the TRPV4-/- animals are protected from hypertrophy. But what is really missing is the attempt to show that TRPV4 inhibition by GSK2193874 is protective and/or provides a therapeutic option which is able to reduce pro-inflammatory downstream signaling.

5) In line 65-66/306-307: The authors state here that "TRPV4 has not been reported in association with pressure overload-induced cardiac hypertrophy", but Adapala et al., 2019 published an abstract showing a role of endothelial TRPV4 in coronary microvascular function during TAC-induced hypertrophy. Therefore, this statement is not correct, and it would be recommendable to change this statement and to expand the discussion/introduction on endothelial-mediated effects in this model.

Adapala, R., Kanugula, A.K., Ohanyan, V.A., Paruchuri, S.M., Chilian, W.M., Thodeti, C.K., 2019. Abstract 15843: Endothelial TRPV4 Deletion Protects Myocardium Against Pressure Overload Induced Hypertrophy via Preserved Angiogenesis and Reduced Cardiac Fibrosis. Circulation 140, A15843-A15843. https://doi.org/10.1161/circ.140.suppl_1.15843

6) The authors refer in the discussion to many studies showing upregulation of TRPV4 (line 302-304) in models of sterile inflammation, but for the readership it is not quite clear what is meant with upregulation (transcriptional or translational level)? Moreover, the references in this passage should be updated, since some of them (Morine et al., 2016  no significant changes; Adapala et al., 2016  lower expression in tumor endothelial cells as compared to normal endothelial cells, but no expression data on GSK790A treatment) are not providing data/supporting the statement of an upregulation. The authors should state more precisely, if they mean activation or expression and on which level. This is critical because many studies show that TRPV4 changes are limited at the mRNA or protein level and that the amount of active TRPV4 channels at the plasma membrane is tightly regulated by vesicular transport.

7) Please provide more information and relevant clinical data for the donors of human samples.

---

## [Author Response]

Essential revisions:1) To examine if TRPV4 plays a role in cardiac hypertrophy, the authors measured the TRPV4 protein and mRNA expression levels in LV tissue from WT and sham-operated BL/6 mice at a single time point of four weeks after TAC surgery. However, it has been demonstrated that after the period longer than two weeks the hearts of TAC operated BL/6 mice aged 10 or 12 weeks manifested cardiac dysfunction by progressing from hypertrophy to heart failure when additional signaling pathways become activated compared to LVH only. This is also indicated by the findings of this study showing "the progression of hypertrophy, inflammation, fibrosis, and heart failure" (p. 21, line 315). Thus, there is no clear distinction between the TRPV4 involvement in development of LVH vs heart failure. This is further addressed in point 3 below.

Thank you very much for the excellent and professional review of our manuscript.

Based on your suggestion, we performed the additional experiments on day 2, day 7, and day 14 after TAC. The protein and mRNA levels of TRPV4 were measured at different time points (2, 7, 14, 28 days) after surgery. We have added the result in the new Figures 1 A-C. We have also added the new Table 1 in Appendix 1, which has listed HW/BW, HW/TL, EF, FS, and the mRNA expression of ANP, BNP, Collagen I, and Collagen III after TAC in WT mice.

On pages 4-5, lines 82-90, we now write:

“Consistent with the previous observation (Guo, et al.,2021), we did not detect significant cardiac hypertrophy on day 2 after TAC. The 1-week TAC time point gave a sample in which the heart was undergoing compensated cardiac hypertrophy, while the 2- and 4-week TAC showed signs of decompensated cardiac hypertrophy with heart failure (Appendix 1-table 1). Similar findings have been previously reported (Wu, et al.,2017). Nonetheless, there are conflicting results for changes in cardiac function, particularly at 2 weeks after TAC, likely caused by differences in severity of constriction. As shown in Figures 1A-C, figure 1-supplement 1, the protein and mRNA level of TRPV4 began to increase 1 week after TAC and maintained a higher level on week 4.”

To further investigate the role of TRPV4 in cardiac hypertrophy, WT and TRPV4-/- mice were subjected to either TAC or sham operation and hypertrophic responses were evaluated at 1 week. We have added the result in the new Figures 2 A-C and revised the second paragraph of the Results section.

On pages 9-10, lines 129-149, we now write:

“To further investigate the role of TRPV4 in cardiac hypertrophy induced by pressure overload, we performed TAC or sham surgery in WT and TRPV4 knock-out (Trpv KO) mice. The hypertrophic response was evaluated 1 week and 4 weeks after TAC. We used the ratios of heart weight/body weight (HW/BW) and heart weight /tibial length (HW/TL) to assess changes in LV mass (Figure 2A). As expected, both values were significantly increased after TAC in WT mice. However, this hypertrophic response to TAC was attenuated in Trpv4 KO mice at 1 week or 4 weeks. Next, we measured the cross-sectional area of myocytes in all groups. As shown in Figure 2B, Trpv4 KO mice significantly attenuated TAC-induced enlargement of myocytes size 1 week or 4 weeks after TAC. In order to confirm our findings at the molecular level, we then determined cardiac hypertrophic marker genes expression. Both ANP (Nppa) and BNP (Nppb) mRNA expression were significantly higher in WT hearts compared with Trpv4 KO hearts 1 week or 4 weeks after TAC. There was no significant difference between WT and Trpv4 KO in the sham group (Figure 2C). These results suggest that TRPV4 activation plays a critical role in pressure overload-induced cardiac hypertrophy.”

We further examined the activation of CaMKII at 2 days, 1 week, 2 weeks, and 4 weeks after TAC in WT mice. We have added the result in the new Figures 8 G-H.

On page 28, lines 366-373, we now write:

“We also examined the phosphorylation of CaMKII at 2 days, 1 week, 2 weeks, and 4 weeks after TAC in WT mice. Similar to our observation of TRPV4 expression, p-CaMKII began to increase 1 week after the operation and remained at a high level 4 weeks after TAC (Figures 8G-H, figure 8 -supplement 3). More interestingly, the TAC-induced upregulation of p-CaMKII in WT mice 1 or 4 weeks was not observed in Trpv4 KO mice (Figures 8I-J, figure 8 -supplement 4). Our results indicate that TRPV4 activation is required for the phosphorylation of CaMKII in response to pressure overload.”

On page 39, lines 502-504 in the Discussion subsection, we now write:

“In addition, our in vivo studies showed that the phosphorylation of CaMKII began to increase at 1 week and maintained higher levels 4 weeks after TAC, which was following the same trend of TRPV4 upregulation.”.

In conclusion, TRPV4 began to increase at 1 week and maintained a higher level 4 weeks after TAC. TRPV4-/- mice, compared with wild type, resulted in significantly reduced cardiac hypertrophy 1 week or 4 weeks after TAC and had improved heart function 4 weeks after TAC. Our results suggest that TRPV4 activation plays a critical role in pressure overload-induced cardiac hypertrophy and heart failure.

2) It is unclear which type of "mechanical loading" (p. 21, line 299) is activating TRPV4. Given the authors' suggestion that TRPV4 activation in pathological cardiac hypertrophy and heart failure occurs in response to pressure overload, does it mean that TRPV4 is activated by membrane stretch? However, this is unlikely because mammalian TRP ion channels are not activated by membrane stretch (Nikolaev et al., J Cell Sci 2019). Consequently, TRPV4 may not play a role as an initiating stressor but is more likely functioning as an upstream signaling transducer (p. 21, line 316). What is then acting upstream of TRPV4 as the primary transducer of the pressure overload? The current study does not answer this relevant question.

Conventionally, the pathological mechanical load has been categorized into pressure- and volume-overload. TAC induces pressure overload and changes the stretch tension of cell membrane. TRPV4 has been found to be directly activated by membrane stretch in *Xenopus laevis* oocytes (Loukin, et al.*,*2010). However, membrane stretch fails to open mammalian TRPV4 (Nikolaev, et al.*,*2019). Therefore, TRPV4 could be a downstream signal initiated by a primary mechanoreceptor, such as Piezo1. Following membrane stretch, Piezo1 is activated, causing a transient calcium influx as well as increased phospholipase A2 (PLA2) activity. PLA2-dependent arachidonic acid metabolites then activate TRPV4, which is observed after the osmotic stimulus and shear stress (Gorelick and Nathanson*,*2020; Watanabe, et al.*,*2003). Indeed, very recent studies have provided evidence that Piezo1 acts as the primary mechanoreceptor and initiates the cardiac hypertrophic response to pressure overload (Guo, et al.*,*2021; Yu, et al.*,*2021). However, further studies are required to investigate a tight interaction of TRPV4 with Piezo1 in pathological cardiac hypertrophy.

The above information has been added in lines 475-486.

3) Yu et al. (Front. Cell Dev. Biol. 2021) reported two ca^2+^-calmodulin dependent hypertrophic pathways leading to LVH; the first one via AngII acting on Gq-coupled receptors and the second one via TAC-induced pressure overload. Both pathways are ca^2+^-calmodulin dependent with the AngII one via the ca^2+^-calmodulin dependent activation of calcineurin and the TAC one via ca^2+^ -calmodulin dependent activation of CaMKII. What is obscuring the results of this study is that the authors use AngII/PE model of cardiac dysfunction (e.g. p. 22, lines 327-32), which although exhibiting hypertrophy, is unrelated to the TAC-induced LVH. Consequently, the signaling pathways are different, and it remains unclear how TRPV4 is activated in this case. Further complication with the phenylephrine is that in addition to acting as a α1-adrenoceptor agonist it also causes ca^2+^ store depletion leading to ca^2+^-induced calcium release from SR through TRPV4 channels (Dryn et al., Int. J. Physiol. Pathophysiol. 2017).

Ang II can potently induce cardiac hypertrophy both in vivo and in vitro. The calcineurin/NFAT pathway has been identified as a likely mechanism (Yu, et al.,2021). However, Ang II also promotes the phosphorylation of CaMKII and increases CaMKII activity in cardiomyocytes (Nakamura and Sadoshima,2018). Similar to our finding, Yang et al (Yang, et al.,2021) have shown that Ang II treatment significantly increases CaMKII phosphorylation in cultured neonatal rat ventricular myocytes (NRVMs). CaMKII induces phosphorylation of the class II histone deacetylase 4 (HDAC4), promotes HDAC4 nuclear report, and then causes cardiac hypertrophy. This pathway has been demonstrated to mediate TAC-induced cardiac hypertrophy (Yu, et al.,2021). Interestingly, pretreatment with KN-93 (a potent CaMKII inhibitor) prevents the Ang II induced-nuclear report of HDAC4 (Helmstadter, et al.,2021). Therefore, the CaMKII pathway might also involve in Ang II-induced cardiac remodeling.

Ang II increases reactive oxygen species (ROS) production and subsequently enhances CaMKII activity (Palomeque, et al.,2009). In a previous study, we reported that TRPV4 activation triggers ca^2+^ entry and then increases ROS production in cardiomyocytes (Wu, et al.,2017). In the present study, we found that pretreatment with TRPV4 antagonist prevented Ang II-induced CaMKII activation. Therefore, we assumed that TRPV4 activation was required for Ang II-induced CaMKII activation. Previous studies have also reported that Ang II can enhance TRPV4 activity and increase TRPV4 expression, through the activation of the AT1 receptor involving the PKC and Src kinase pathway, respectively (Mercado, et al.,2014; Saxena, et al.,2014). However, future studies addressing the cross-talk between TRPV4 and AngII in the context of cardiac hypertension are needed.

Similar to Ang II, PE can also promote cardiac hypertrophy both in vivo and in vitro. PE, the α-adrenergic receptor agonist, can activate the Gq/11 subfamily of G proteins, leading to the activation of phospholipase C (PLC). The PLC hydrolysis of phosphatidylinositol 4,5-bisphosphate produces two second messengers, inositol 1,4,5-trisphosphate (IP3) and 1,2-diacylglycerol (DAG). IP3 has been found to increase ca^2+^ influx via TRPV4 (Garcia-Elias, et al.,2008; Hong, et al.,2018; Takahashi, et al.,2014). In addition, PE may increase TRPV4 activity via the DAG/PKC signal pathway. Many previous studies have shown that PE increases the phosphorylation of CaMKII and subsequently induces cardiac hypertrophy (Ji, et al.,2017; Ramirez, et al.,1997; Tonegawa, et al.,2017).

On lines 463-467, we now write:

“Previous studies have reported that Ang II can enhance TRPV4 activity and increase TRPV4 expression, through the activation of AT1 receptor involving the PKC and Src kinase pathway, respectively (Mercado, et al.,2014; Saxena, et al.,2014). Future studies addressing TRPV4 upregulation in the context of cardiac hypertrophy are needed.”

On lines 498-502, we now write:

“Previous studies have also shown that Ang II/PE promotes the phosphorylation of CaMKII, increases CaMKII activity, and subsequently mediates cardiac hypertrophy (Helmstadter, et al.,2021; Nakamura and Sadoshima,2018; Tonegawa, et al.,2017). However, the calcineurin/NFAT pathway may play a more important role in AngII-induced cardiac hypertrophy (Yu, et al.,2021).”

4) The data of the TAC model are convincing, and the TRPV4-/- animals are protected from hypertrophy. But what is really missing is the attempt to show that TRPV4 inhibition by GSK2193874 is protective and/or provides a therapeutic option which is able to reduce pro-inflammatory downstream signaling.

As you suggested, we performed an additional experiment to investigate the therapeutic role of the TRPV4 antagonist GSK2193874 in TAC-induced hypertrophy and heart failure. As expected, GSK2193874 protected against pressure overload-induced cardiac hypertrophy as evidenced by measuring the mice phenotype, including global heart size, HW/BW ratio, HW/TL ratio, histological examinations of hypertrophy, fibrosis, echocardiographic measurements of systolic function, and mRNA levels of hypertrophic markers. (new Figure 5).

The new results have been added in the Results section on pages 20-21. The figures appear in the new Figure 5. The corresponding figure legend has been also added.

We have added the text, “Furthermore, the TRPV4 antagonist inhibited cardiac remodeling and dysfunction induced by TAC.” in the Abstract.

On pages20-21, lines 259-274, we now write:

“TRPV4 antagonist attenuates the pathological cardiac remodeling induced by TAC

We further evaluated the effects of TRPV4 inhibition with a specific antagonist GSK2193874 (GSK3874) on pathological cardiac hypertrophy and dysfunction induced by pressure overload. WT mice received intragastric administration of GSK3874 at a dose of 10 mg/kg/d or vehicle from the day of TAC operation (Thorneloe, et al.,2012; Liao, et al.,2020). After 4 weeks, GSK3874 treatment substantially suppressed the increase in heart size induced by TAC (Figure 5A). Moreover, HW/BW and HW/TL ratios significantly decreased in GSK3874-treated mice (Figures 5B-C). The cross-sectional area of cardiomyocytes was also markedly reduced after treatment with GSK3874 (Figure 5D). The mRNA expression of hypertrophy marker genes ANP (Nppa) and BNP (Nppb) was significantly reduced (Figures 5E-F). Echocardiography showed that EF% and FS% were also significantly higher in GSK3874- than in the vehicle-treated group (Figures 5 G-H). LV internal dimension systole and LV mass were significantly smaller following GSK3874 treatment (Figures 5I-J). Finally, the fibrosis area significantly decreased in GSK3874-treated mice (Figure 5K). In summary, these results indicate that GSK3874 treatment effectively reduces cardiac remodeling and dysfunction induced by TAC.”

On page 48, lines 598-600, we now write:

“WT mice were treated with vehicle (6% Cavitron) or GSK3874 (10 mg/kg/d) via oral gavage for 4 weeks after TAC (Liao, et al.,2020; Thorneloe, et al.,2012).”

5) In line 65-66/306-307: The authors state here that "TRPV4 has not been reported in association with pressure overload-induced cardiac hypertrophy", but Adapala et al., 2019 published an abstract showing a role of endothelial TRPV4 in coronary microvascular function during TAC-induced hypertrophy. Therefore, this statement is not correct, and it would be recommendable to change this statement and to expand the discussion/introduction on endothelial-mediated effects in this model.Adapala, R., Kanugula, A.K., Ohanyan, V.A., Paruchuri, S.M., Chilian, W.M., Thodeti, C.K., 2019. Abstract 15843: Endothelial TRPV4 Deletion Protects Myocardium Against Pressure Overload Induced Hypertrophy via Preserved Angiogenesis and Reduced Cardiac Fibrosis. Circulation 140, A15843-A15843. https://doi.org/10.1161/circ.140.suppl_1.15843

We appreciate and agree with the reviewer’s comments. We have added the response statements to the “Introduction” section in this revised manuscript.

On page 4, lines 69-72, we now write:

“Moreover, Adapala et al. have reported that endothelial TRPV4 deletion protects TAC-induced-structure remodeling in a conference abstract (Adapala, et al.,2019). However, the role of TRPV4 in the development of pressure overload-induced cardiac hypertrophy is not well understood.”.

We also deleted the previous statement “However, there are no data demonstrating the role of TRPV4 in pathological cardiac hypertrophy and heart failure in response to pressure overload.”.

In the “Discussion” section, on page 40, lines 531-533, we now write:

“Indeed, deletion of endothelial TRPV4 has been found to suppress TAC-induced cardiac hypertrophy and dysfunction via increased coronary angiogenesis and reduced cardiac fibrosis (Adapala, et al.,2019)”.

Thank you for your comment.

6) The authors refer in the discussion to many studies showing upregulation of TRPV4 (line 302-304) in models of sterile inflammation, but for the readership it is not quite clear what is meant with upregulation (transcriptional or translational level)? Moreover, the references in this passage should be updated, since some of them (Morine et al., 2016  no significant changes; Adapala et al., 2016  lower expression in tumor endothelial cells as compared to normal endothelial cells, but no expression data on GSK790A treatment) are not providing data/supporting the statement of an upregulation. The authors should state more precisely, if they mean activation or expression and on which level. This is critical because many studies show that TRPV4 changes are limited at the mRNA or protein level and that the amount of active TRPV4 channels at the plasma membrane is tightly regulated by vesicular transport.

As you suggested, we have revised the second paragraph of the “Discussion” section. Please check page 37, line 453.

Morine et al., 2016 have shown that the mRNA level of TRPV4 increases more than 8-fold 8 weeks after thoracic aorta constriction in Eng^+/+^ mice (please check their Figure 3B). Adapala et al. have found that TRPV4 protein level in tumor endothelial cells is lower than normal endothelial cells but can be significantly increased (more than 1.5 -fold) after 24 h treatment with TRPV4 agonist GSK790A (please check their Figure 4 C). Indeed, several other groups have also demonstrated that TRPV4 function is enhanced even if its expression remains unchanged under pathological conditions (Everaerts, et al.*,*2010; Rahaman, et al.*,*2014)

7) Please provide more information and relevant clinical data for the donors of human samples.

As you suggested, the clinical data of patients have been shown in Appendix 2-table 1.

On page 45, lines 549-553, we now write:

“All patients were diagnosed with DCM with ejection fraction (EF) less than 40% at least 3 months prior to heart transplantation. Non-heart failure tissues were obtained from three organ donors whose hearts could not be placed due to size issues, ABO mismatch, or other factors. The clinical data of patients have been shown in Appendix 2-table 1. The clinical data of patients have been shown in Appendix 2-table 1.”.

References:

Everaerts W, Zhen X, Ghosh D, Vriens J, Gevaert T, Gilbert JP, Hayward NJ, McNamara CR, Xue F, Moran MM, Strassmaier T, Uykal E, Owsianik G, Vennekens R, De Ridder D, Nilius B, Fanger CM, Voets T. 2010. Inhibition of the cation channel TRPV4 improves bladder function in mice and rats with cyclophosphamide-induced cystitis. *Proc Natl Acad Sci U S A* 107:19084-19089. DOI:10.1073/pnas.1005333107, PMID:20956320

Garcia-Elias A, Lorenzo IM, Vicente R, Valverde MA. 2008. IP3 receptor binds to and sensitizes TRPV4 channel to osmotic stimuli via a calmodulin-binding site. *J Biol Chem* 283:31284-31288. DOI:10.1074/jbc.C800184200, PMID:18826956

Gorelick F, Nathanson MH. 2020. TRPV4 helps Piezo1 put the squeeze on pancreatic acinar cells. *J Clin Invest* 130:2199-2201. DOI:10.1172/JCI136525, PMID:32281947

Guo Y, Merten AL, Scholer U, Yu ZY, Cvetkovska J, Fatkin D, Feneley MP, Martinac B, Friedrich O. 2021. in vitro cell stretching technology (IsoStretcher) as an approach to unravel Piezo1-mediated cardiac mechanotransduction. *Prog Biophys Mol Biol* 159:22-33. DOI:10.1016/j.pbiomolbio.2020.07.003, PMID:32763257

Helmstadter KG, Ljubojevic-Holzer S, Wood BM, Taheri KD, Sedej S, Erickson JR, Bossuyt J, Bers DM. 2021. CaMKII and PKA-dependent phosphorylation co-regulate nuclear localization of HDAC4 in adult cardiomyocytes. *Basic Res Cardiol* 116:11. DOI:10.1007/s00395-021-00850-2, PMID:33590335

Hong K, Cope EL, DeLalio LJ, Marziano C, Isakson BE, Sonkusare SK. 2018. TRPV4 (Transient receptor potential vanilloid 4) Channel-Dependent negative feedback mechanism regulates gq Protein-Coupled Receptor-Induced vasoconstriction. *Arterioscler Thromb Vasc Biol* 38:542-554. DOI:10.1161/ATVBAHA.117.310038, PMID:29301784

Ji Y, Guo X, Zhang Z, Huang Z, Zhu J, Chen QH, Gui L. 2017. CaMKIIdelta meditates phenylephrine induced cardiomyocyte hypertrophy through store-operated ca^2+^ entry. *Cardiovasc Pathol* 27:9-17. DOI:10.1016/j.carpath.2016.11.004, PMID:27940402

Loukin S, Zhou X, Su Z, Saimi Y, Kung C. 2010. Wild-type and brachyolmia-causing mutant TRPV4 channels respond directly to stretch force. *J Biol Chem* 285:27176-27181. DOI:10.1074/jbc.M110.143370, PMID:20605796

Mercado J, Baylie R, Navedo MF, Yuan C, Scott JD, Nelson MT, Brayden JE, Santana LF. 2014. Local control of TRPV4 channels by AKAP150-targeted PKC in arterial smooth muscle. *J Gen Physiol* 143:559-575. DOI:10.1085/jgp.201311050, PMID:24778429

Nakamura M, Sadoshima J. 2018. Mechanisms of physiological and pathological cardiac hypertrophy. *Nat Rev Cardiol* 15:387-407. DOI:10.1038/s41569-018-0007-y, PMID:29674714

Nikolaev YA, Cox CD, Ridone P, Rohde PR, Cordero-Morales JF, Vasquez V, Laver DR, Martinac B. 2019. Mammalian TRP ion channels are insensitive to membrane stretch. *J Cell Sci* 132:s238360. DOI:10.1242/jcs.238360, PMID:31722978

Palomeque J, Rueda OV, Sapia L, Valverde CA, Salas M, Petroff MV, Mattiazzi A. 2009. Angiotensin II-induced oxidative stress resets the ca^2+^ dependence of ca^2+^-calmodulin protein kinase II and promotes a death pathway conserved across different species. *Circ Res* 105:1204-1212. DOI:10.1161/CIRCRESAHA.109.204172, PMID:19850941

Rahaman SO, Grove LM, Paruchuri S, Southern BD, Abraham S, Niese KA, Scheraga RG, Ghosh S, Thodeti CK, Zhang DX, Moran MM, Schilling WP, Tschumperlin DJ, Olman MA. 2014. TRPV4 mediates myofibroblast differentiation and pulmonary fibrosis in mice. *J Clin Invest* 124:5225-5238. DOI:10.1172/JCI75331, PMID:25365224

Ramirez MT, Zhao XL, Schulman H, Brown JH. 1997. The nuclear deltaB isoform of ca^2+^/calmodulin-dependent protein kinase II regulates atrial natriuretic factor gene expression in ventricular myocytes. *J Biol Chem* 272:31203-31208. DOI:10.1074/jbc.272.49.31203, PMID:9388275

Saxena A, Bachelor M, Park YH, Carreno FR, Nedungadi TP, Cunningham JT. 2014. Angiotensin II induces membrane trafficking of natively expressed transient receptor potential vanilloid type 4 channels in hypothalamic 4B cells. *Am J Physiol Regul Integr Comp Physiol* 307:R945-R955. DOI:10.1152/ajpregu.00224.2014, PMID:25080500

Takahashi N, Hamada-Nakahara S, Itoh Y, Takemura K, Shimada A, Ueda Y, Kitamata M, Matsuoka R, Hanawa-Suetsugu K, Senju Y, Mori MX, Kiyonaka S, Kohda D, Kitao A, Mori Y, Suetsugu S. 2014. TRPV4 channel activity is modulated by direct interaction of the ankyrin domain to PI(4,5)P. *Nat Commun* 5:4994. DOI:10.1038/ncomms5994, PMID:25256292

Tonegawa K, Otsuka W, Kumagai S, Matsunami S, Hayamizu N, Tanaka S, Moriwaki K, Obana M, Maeda M, Asahi M, Kiyonari H, Fujio Y, Nakayama H. 2017. Caveolae-specific activation loop between CaMKII and L-type Ca(2+) channel aggravates cardiac hypertrophy in alpha1-adrenergic stimulation. *Am J Physiol Heart Circ Physiol* 312:H501-H514. DOI:10.1152/ajpheart.00601.2016, PMID:28039202

Watanabe H, Vriens J, Prenen J, Droogmans G, Voets T, Nilius B. 2003. Anandamide and arachidonic acid use epoxyeicosatrienoic acids to activate TRPV4 channels. *Nature* 424:434-438. DOI:10.1038/nature01807, PMID:12879072

Wu QF, Qian C, Zhao N, Dong Q, Li J, Wang BB, Chen L, Yu L, Han B, Du YM, Liao YH. 2017. Activation of transient receptor potential vanilloid 4 involves in hypoxia/reoxygenation injury in cardiomyocytes. *Cell Death Dis* 8:e2828. DOI:10.1038/cddis.2017.227, PMID:28542130

Yang B, Jiang Q, He S, Li T, Ou X, Chen T, Fan X, Jiang F, Zeng X, Huang CL, Lei M, Tan X. 2021. Ventricular SK2 upregulation following angiotensin II challenge: Modulation by p21-activated kinase-1. *J Mol Cell Cardiol* 164:110-125. DOI:10.1016/j.yjmcc.2021.11.001, PMID:34774547

Yu Z, Gong H, Kesteven S, Guo Y, Wu J, Li J, Iismaa S, Kaidonis X, Graham R, Cox C. 2021. Piezo1 is the cardiac mechanosensor that initiates the hypertrophic response to pressure overload. *Research Square* DOI:10.21203/rs.3.rs-895561/v1,

Yu ZY, Gong H, Wu J, Dai Y, Kesteven SH, Fatkin D, Martinac B, Graham RM, Feneley MP. 2021. Cardiac gq receptors and calcineurin activation are not required for the hypertrophic response to mechanical left ventricular pressure overload. *Front Cell Dev Biol* 9:639509. DOI:10.3389/fcell.2021.639509, PMID:33659256